# Quantitation of ethanol in UTI assay for volatile organic compound detection by electronic nose using the validated headspace GC-MS method

**Nam Than**[1¤], **Zamri Chik**[2], **Amy Bowers**[3], **Luisa Bozano**[3], **Aminat Adebiyi**[3]*

1 Department of Biomedical Engineering, San Jose State University, San Jose, California, United States of America, 2 Universiti Malaya Bioequivalence Testing Centre (UBAT), Department of Pharmacology, Faculty of Medicine, Universiti Malaya, Kuala Lumpur, Malaysia, 3 IBM Almaden Research Center, San Jose, California, United States of America

☯ These authors contributed equally to this work.
¤ Current address: Department of Biomedical Engineering, The University of Texas at Austin, Austin, Texas, United States of America
* aminat.adebiyi@ibm.com

**Data Availability Statement:** The data files are available on Kaggle with unrestricted access: - GCMS data – raw files used to develop validation table - https://www.kaggle.com/datasets/aminatadebiyi/beva-plos-one - IBM EVA data and

## Abstract

Disease detection through gas analysis has long been the topic of many studies because of its potential as a rapid diagnostic technique. In particular, the pathogens that cause urinary tract infection (UTI) have been shown to generate different profiles of volatile organic compounds, thus enabling the discrimination of causative agents using an electronic nose. While past studies have performed data collection on either agar culture or jellified urine culture, this study measures the headspace volume of liquid urine culture samples. Evaporation of the liquid and the presence of background compounds during electronic nose (e-nose) device operation could introduce variability to the collected data. Therefore, a headspace gas chromatography-mass spectrometry method was developed and validated for quantitating ethanol in the headspace of the urine samples. By leveraging the new method to characterize the sample stability during e-nose measurement, it was revealed that ethanol concentration dropped more than 15% after only three measurement cycles, which equal 30 minutes for this study. It was further shown that by using only data within the first three cycles, better accuracies for between-day classification were achieved, which was 73.7% and 97.0%, compared to using data from within the first nine cycles, which resulted in 65.0% and 81.1% accuracies. Therefore, the newly developed method provides better quality control for data collection, paving ways for the future establishment of a training data library for UTI.

## Introduction

It is widely known that different types of disease can influence the pattern of volatile organic biomarkers released in exhaled gas or waste materials such as fecal matter or urine [1–3]. This is especially true for infectious diseases, stemming from the fact that different bacterial species

features for Artificial Neural Network - https://www.kaggle.com/datasets/aminatadebiyi/beva-plos-one-ann.

**Funding:** NT completed this research as part of graduate studies with San Jose State University. ZC participated with financial grant support from the Ministry of Higher Education Malaysia as part of the Sub-Specialty Training Program. IBM Research provided support in the form of salary for AA, AB & LB, and material funding for experiments, but did not have any additional role in the study design, data collection and analysis, decision to publish, or preparation of the manuscript.

**Competing interests:** The authors have declared that no competing interests exist.

can generate different profiles of volatile organic compounds (VOCs). Indeed, Bos et al. systematically reviewed the literature for the headspace VOC of six bacterial species and found that while they share some compounds, they form different patterns as some compounds are uniquely produced by only certain species [4]. Even among compounds that are commonly produced by multiple species, a study using Selected Ion Flow Tube Mass Spectrometry (SIFT-MS) revealed that the produced concentrations are different, thus still ensuring the unique pattern for species differentiation [5]. These fingerprint VOC profiles can be examined with headspace analysis and classified using a computer algorithm to enable remarkable detection of urinary tract infection [3, 6]. A recent review by Dospinescu et al. on studies of UTI-associated bacterial VOCs revealed a heavy reliance on Gas-liquid chromatography to discriminate VOC patterns until recently when e-noses emerged as a promising technology for identifying infectious UTI-causing strains in a clinical setting [7]. The use of the e-nose as a rapid diagnostic tool is valuable for clinical use because the gold standard of UTI detection is bacterial culture which can take days to obtain a result. While a dipstick is also a rapid, fairly accurate, and low-cost candidate, a key advantage of the e-nose is the potential to identify specific bacterial species in real-time.

Static headspace gas chromatography is a technique used for the concentration analysis of volatile organic compounds. This technique is relatively simple and can provide sensitivity that is similar to dynamic purge and trap analysis. The popularity of this technique has grown and gained worldwide acceptance for analyses of alcohol in blood, urine, and other biological samples, as well as residual solvents in pharmaceutical products. Sample matrices like blood, plastic, and cosmetics contain high molecular weights, non-volatile material that can remain in the GC system and result in poor analytical performance. Many laboratory analysts use extensive sample preparation techniques to extract and concentrate the compounds of interest from this unwanted non-volatile material. These extraction and concentration techniques can become time-consuming and costly. Static headspace analysis avoids this time and cost by directly sampling the volatile headspace from the container in which the sample is placed.

While state-of-the-art methods such as GC-MS and SIFT-MS are useful in characterizing VOC profiles, they are usually expensive, non-portable, and time-consuming in terms of sample preparation. As alternatives, there are a plethora of studies using commercially available sensors for the same purpose, which led to a technology called Electronic Nose (e-nose) [8]. An e-nose is essentially a system of gas sensors with pattern recognition capability to detect diseases. An e-nose must consist of appropriate hardware and software. Hardware refers to the type of sensors, most commercially available being Metal Oxide (MOX), and software refers to the analytical method used, usually in the form of a machine learning algorithm. Table 1 summarizes some different combinations of sensors and analytical methods used in past studies. Noticeably, a minimum of six sensors with an artificial neural network (ANN) model was enough to yield a high prediction accuracy, albeit a small number of labels of two.

While many more e-nose technologies exist for other applications, Table 1 is a survey of urine-based or pathogen detection studies. Many of these studies were conducted using non-physiological media such as nutrient broth, culture media, agar, or urine made into an agar. On the other hand, many urine-based studies did not involve pathogen classification in the context of UTI. Using LDA, PCA, and cluster analysis, many authors aimed to discriminate between only two labels, healthy controls and diseased samples for a variety of diseases, such as prostate cancer [13, 16], bacteriuria [14], diabetes mellitus [15, 19], and Azotemia [18]. We envisioned that point-of-care prediction of UTI should be performed directly on a patient's urine sample with more labels to identify a wider range of causative agents. This combined lack of e-nose studies on liquid urine and a more complex classification model presents a need

**Table 1. Literature survey of sensors and classification models for e-nose studies.**

| Study | Sensors | Analysis | Labels | Medium | Average Accuracy |
|---|---|---|---|---|---|
| Craven (1997) [9] | 6 MOX | Multi-layer perceptron and Linear discriminant analysis (LDA) | 4 | Nutrient broth | 82.2% |
| Gibson et al. (1997) [10] | 14 Conductive polymers | Multi-layer perceptron | 13 | Nutrient agar | 89.7% |
| Gardner et al. (2000) [11] | 6 MOX | ANN with backpropagation | 2 | Blood agar then nutrient broth | 96.0% |
| Pavlou et al. (2002) [12] | 14 Conductive polymers | Genetic algorithm and backpropagation neural network | 4 | Agar, Brain heart infusion, cooked meat broth | 95.0% |
| Roine et al. (2014) [6] | Commercial ion mobility spectrometer-based e-nose and 6 MOX | LDA and logistic regression | 5 | Normal urine made into an agar | 83.9% |
| Asimakopoulos (2014) [13] | 8 metalloporphyrin-coated sensors | Supervised Partial Least Square–Discriminant Analysis | 2 | Urine | 84.8% |
| Aathithan et al. (2001) [14] | 4 conductive polymers | Principal component analysis (PCA) | 2 | Artificial urine | 72.30% sensitivity; 89.38% specificity |
| Seesaard et al. (2016) [15] | 4 nanocomposites | PCA and cluster analysis | 2 | Urine | 99.5% |
| Filianoti et al. (2022) [16] | Cyranose 320 | Linear canonical discriminant analysis | 2 | Urine | 85.3% |
| Seesaard et al. (2020) [17] | A hybrid of 3 nanocomposites and 3 MOX | PCA and cluster analysis | 4 | Bacterial culture media | 99.7% |
| Yumang et al. (2020) [18] | 7 MOX | PCA then K-nearest neighbor analysis | 2 | Urine | 90% |
| Esfahani et al. (2018) [19] | Field-Asymmetric Ion Mobility Spectrometry (FAIMS) or FOX4000 (18 MOX) | Sparse Logistic Regression, Random Forest, Gaussian Process, and Support Vector | 2 | Urine | 85–94%, depending on e-nose choice and sample age |

for urine-based tests for UTI on an e-nose equipped with a more powerful classification model.

*Escherichia coli* is known to be the most prevalent cause of UTI [20]. Therefore, a simple *in vitro* UTI model could be created by infecting commercially available human urine with pathogenic *E. coli* to test the e-nose system. Ethanol is a by-product of *E. coli* metabolism with lactose and arabinose and was considered a biomarker for *E. coli*, and ethanol level was also indicative of bacterial concentration in the sample [7]. However, urine samples are known to degrade over time [21, 22]. Volatile urine output has been shown to decrease after nine months of storage at -80°C [23]. Urine analysis performed on samples stored within a year resulted in much better accuracy than including samples within four years [19]. Therefore, minimization of the variability in the collected data caused by evaporation and sample degradation must be achieved before confidently and correctly establishing a library of training data for future real-time, point-of-care prediction of UTI. Therefore, a GC-MS method was needed for characterizing the headspace samples so that decisions can be made regarding the optimal measurement time, followed by the exclusion of examples that do not genuinely represent the intended labels, which could poison the analytical model.

There was little evidence in the literature that there exists an exact method for the determination of ethanol concentration in urine using headspace GC-MS. Table 2 summarizes the literature survey. Also, Tangerman argued that the headspace technique is notorious for requiring a substantial amount of labor and volumes of the biological specimen while being less sensitive [24]. However, the headspace method more closely represents the approach done with the e-noses. Therefore, a new method for the quantitation of ethanol concentration in the

**Table 2. Literature survey of gc methods for the quantitation of ethanol in biological matrices.**

| Study | Analyte | Sample Matrix | Method |
|---|---|---|---|
| Mihretu et al. (2020) [25] | Ethanol | Blood | Headspace GC-FID (Flame ionization detector) |
| Chun et al. (2016) [26] | Alcohols | Brain tissue | Headspace GC-FID |
| Xiao et al. (2014) [27] | Ethanol | Blood | Headspace GC-MS |
| Kristoffersen et al. (2006) [28] | Ethanol | Whole blood and plasma | Headspace GC-FID |
| Smith et al. (1999) [29] | Alcohols | Urine | Headspace SIFT-MS |
| Tangerman (1997) [24] | Ethanol | Whole blood, serum, urine, fecal supernatants | Direct Injection GC-MS |

headspace volume of urine samples inoculated with *E. coli* was developed and validated in this study. Finally, it was used to help improve the classification accuracy of the EVA.

There are also portable systems capable of detecting and quantifying ethanol, such as zNose, or GC-IMS, which are gas sensors coupled with a GC column, or an alcohol breath analyzer, which employs a single optical gas sensor. However, we are interested in quantifying ethanol to assist with establishing a working protocol for the EVA. Therefore, a GC-MS method was developed and validated for ethanol quantification in urine. To our knowledge, this is the first paper describing the validation and quantitation of alcohol release from the bacteria by using inoculated urine samples. Using the method, we devised a plan to determine the optimal time length for measuring the urine samples to maximize the data collected without sacrificing classification accuracy.

In this paper, we use the Electronic Volatile Analyzer (EVA), a rapid, gas-sensing Internet of Things (IoT) platform under development at IBM Research, Almaden–designed to deliver classification results in under two minutes [30]. The main objective of this study was to demonstrate the utility of EVA in differentiating between a nonpathogenic (K12) and a uropathogenic (UPEC) strain of *E. coli* in inoculated urine and the possibility of classification improvement by using a validated GC-MS method to quantify ethanol reduction as a sign of diminished sample quality.

## Materials and methods

### The IBM electronic volatile analyzer™

The EVA (Fig 1) an electronic nose under development at IBM Research Almaden. The platform consists of a modular design with each gas sensor mounted on its own sensor module which is a printed circuit board of common design carrying an integrated microcontroller and the circuitry needed to operate the sensor. The sensor modules communicate via I2C protocol with a central hub (BeagleBone Black), which is a single-board computer that orchestrates the modules and processes the multi-sensorial output. Table 3 describes the set of six commercial MOX gas sensors used to collect the measurements described in the following sections. The selection of the sensor portfolio was based on the indications of their respective manufacturers regarding the nominal target gases of each sensor. It is imperative to include sensors for various compounds to capture enough differential responses for pattern recognition of complex VOC profiles. These sensors together target CO, $H_2$, ethanol, methane, ammonia, $H_2S$, and unspecified combustible gases. Consequently, the as-selected sensor array was expected to respond to a variety of volatile molecules, including alcohols, hydrocarbons, ammonia, methane, as well as a wide range of volatile organic compounds (VOC), with a lower limit of detection at parts per million (ppm) level.

Each MOX sensor was operated using an individualized multi-step, periodic voltage profile applied to the heating element, which resulted in stepwise modulation of the device

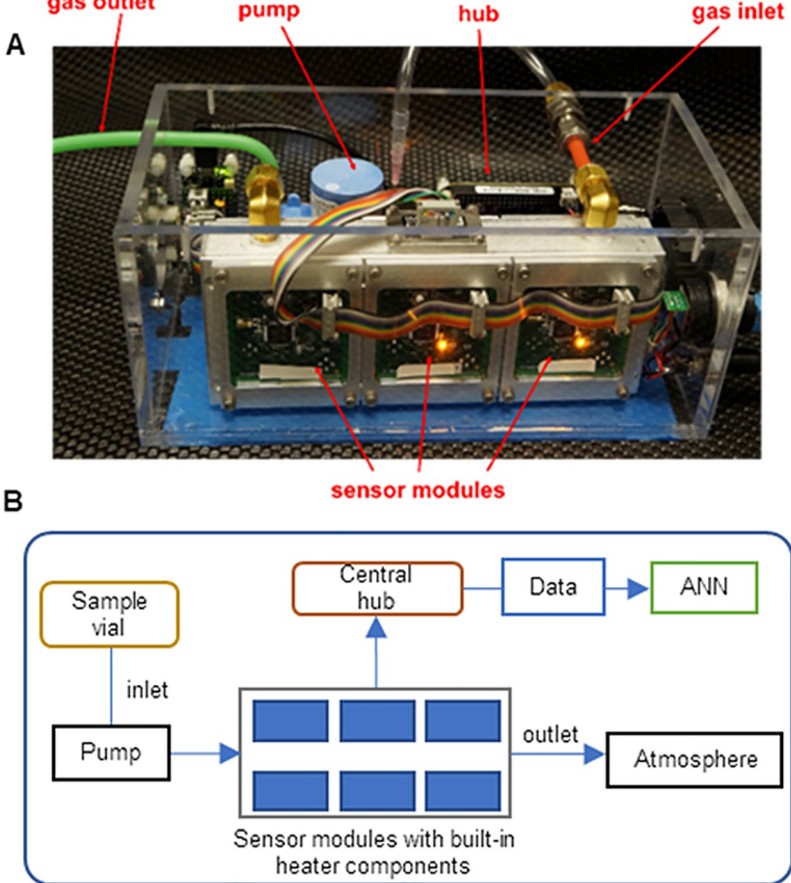

**Fig 1. The Electronic Volatile Analyzer (EVA).** (A) Sensor array prototype with parts labeled. (B) A block diagram of key components of the EVA.

temperature (Fig 2). Modulation of the temperature of MOX sensing elements is a well-known technique that can be used to enhance and tune the dependence of the sensing element resistance to the surrounding environment. The heater voltage profiles applied to the EVA sensors were optimized to maximize the sensitivity of each sensor with respect to a subset of target VOC, as well as to improve response orthogonality. Although the duration and amplitude of the individual waveforms was adjusted independently for each sensor, all waveforms were synchronized to a period of 80 s to simplify the handling and processing of the sensor array outputs. The resistance of each MOX sensing element was monitored at a fixed voltage and a rate

**Table 3. MOX sensors for the electronic volatile analyzer.**

| Sensor | Targeted gases | Commercial application | Manufacturer |
|---|---|---|---|
| GGS2330 | CO, $H_2$, ethanol | Wide range applications | Umwelt Sensor Technik, Germany |
| GGS1330 | Hydrocarbon, combustible gases | Gas leak detection | Umwelt Sensor Technik, Germany |
| TGS2611 | Methane | Gas leak detection | Figaro USA, Inc., USA |
| TGS2602 | VOCs and odorous gases such as ammonia and $H_2S$ | Indoor air quality monitoring | Figaro USA, Inc., USA |
| TGS2600 | $H_2$, ethanol, air pollutants | Indoor air quality monitoring | Figaro USA, Inc., USA |
| TGS8100 | H2, ethanol, air pollutants | Indoor air quality monitoring | Figaro USA, Inc., USA |

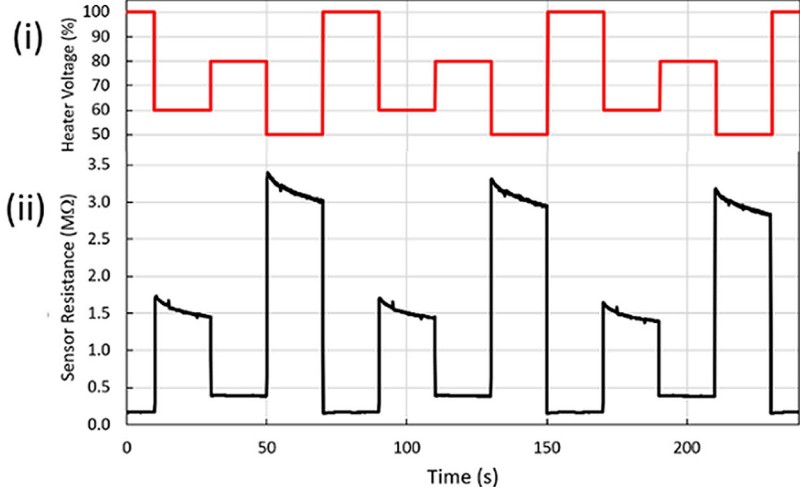

**Fig 2. Example of a temperature profile modulation for MOX sensors for IBM EVA™: (i) periodic waveform of heater voltage, expressed as a percentage of the maximum operating voltage recommended by the sensor manufacturer; (ii) corresponding variations in MOX sensor resistance under constant environment.**

of 10 Hz. During operation, a constant flow of 150 ± 10 sccm was established by means of a small pump, drawing air from the environment in a continuous fashion while also collecting the headspace of the sample of interest. The vapor-carrying flow was directed towards an enclosed chamber containing the six MOX gas sensors for the exposure and detection to take place.

## Chemicals and reagents

Normal Human Urine was purchased from UTAK Laboratories Inc., USA. The vendor obtained consent from the donors, who were healthy and drug-free. Two strains of *E. coli* were purchased from the American Type Culture Collection (ATCC). A nonpathogenic strain (ATCC 29425) is designated as K12, and the other is a uropathogenic strain (ATCC 700928) designated as CFT073. They are, henceforth, referred to as K12 and UPEC, respectively. Bacterial growth medium was prepared from Tryptic Soy Broth (TSB) powder (BD Bacto). For GC-MS, HPLC-grade ethanol (EtOH) and isopropyl alcohol (IPA) were obtained from Sigma Aldrich, USA.

## Preparation of EVA samples and GC-MS reagents

After thawing, the urine bottle was vigorously shaken and aliquoted into multiple 50-ml centrifuge tubes. Upon usage, the tubes were centrifuged at 15,000 rpm for 15 minutes to concentrate any visible sediments, which were then removed by filtration. The resultant urine was named filtered normal urine or fNU.

The TSB medium was mixed and autoclaved using the standard method printed on its bottle. The powder form of the bacteria was resuspended in liquid TSB. Each suspension was then streaked on a Tryptic Soy Agar using standard aseptic techniques. The presence of many colonies after 24 hours of incubation at 37°C indicated bacterial viability. The handling of the UPEC was carried out by BSL-2 trained personnel under a BSL-2 cabinet. The agars with colonies were stored at 4–7°C and re-streaked every two weeks to keep the cell line fresh.

One colony of each *E. coli* strain was aseptically inoculated in separate tubes of 10 ml of fNU. The cultures were incubated for 24 hours before being filtered using the same procedure for fNU, except the centrifugation speed was 5000 rpm to avoid the destruction of the bacterial cells that can inadvertently introduce unwanted substances to the supernatant. Five milliliters of the supernatant were transferred to a septa-top vial and kept in the refrigerator at 4–7˚C until analysis.

A stock solution for EtOH at a 1/100 dilution was prepared by adding 1 ml of pure, 200-proof ethanol in 9 ml of Millipore water in a tube labeled 1/10 EtOH. After vortexing, 1 ml of the 1/10 EtOH was added to 9 ml of Millipore water in a tube labeled 1/100 EtOH and vortexed. The 1/100 solution of IPA was prepared in a similar manner from a bottle of pure IPA. The equivalent concentration in g/ml and ppm for both solutions is 0.008 g/ml and 8 ppm.

## EVA measurement and two-fold cross-validation ANN

Samples were prepared in Wheaton Septa-top Vials with a rubber-top cap that can be punctured with two needles to allow air intake. The first needle is connected to the EVA by a silicon tubing for headspace flux, and the second is connected to a HEPA carbon filter (0.22 μm) and left open to lab air for venting. Four samples were measured with EVA: an empty vial as lab air, 5 ml of normal fNU, 5 ml of K12, and 5 ml of UPEC. Each vial was measured sequentially at a sampling rate of 10 Hz. Additionally, the measurement sequence was randomized to minimize any history effects. Each sample was measured once in each cycle for ten minutes in a continuous flow and reiterated after all other samples have been measured. Between each vial measurement, the device was also flushed for five minutes to purge off residual VOCs.

With temperature oscillation resulting from the heater voltage waveform over 80 seconds, the six sensors give rise to a total of 120 features. Feature extraction was achieved through an amplitude-driven approach of extracting the mean area under the curve for the given response duration using GNU Octave v-5.1.0.0 (GUI), available through open source. Feature extraction every 80s constitutes one training example for the ANN model for a total of 7 examples per sample per cycle. Upon determining the training and testing sets by randomly splitting the overall dataset in half, both datasets are fed into a backpropagating ANN with two hidden layers of 24 and 9 nodes, respectively, which reports the training and testing results. The accuracy was calculated by dividing the total number of correct predictions over the total number of examples. Second-fold cross-validation was performed by switching the training and testing sets and repeating the analytics. The robustness of the ANN model was assessed with both accuracies.

## Preparation of calibration standards

Calibration standards were prepared by spiking urine with certain amounts of EtOH and IPA. The final alcohol concentration was calculated using the following equation:

$$C = \frac{X \cdot D \cdot V_{alc}}{V_{urine}} \cdot \frac{10^6 \, ppm}{1 \, g/ml}$$

where C is the concentration in ppm; X is the dilution factor, which is 1/100; D is the density of the alcohols, which is 0.8 g/ml for both IPA and EtOH; $V_{alc}$ is the volume in μl of the alcohols and $V_{urine}$ the volume in μl of urine.

A preliminary investigation found that the EtOH in UPEC-inoculated urine was approximately 20 to 30 ppm using a non-validated headspace GC-MS analysis with random EtOH calibrators. Based on this result, a more defined range of concentrations for the EtOH calibrators

were determined to be 10, 15, 20, 30, 50, 75, and 100 ppm. The final volume reflects the ultimate amount after transferring some amount to make lower dilutions (e.g., 12.0 ml of the 100-ppm concentration was used to make the 75-ppm concentration). Approximately 16 milliliters of each concentration were prepared so that they could be split into three batches for subsequent inter-day validation. The exact volume for each concentration was carefully calculated and tabulated in Table 4. After aliquoting 5 ml of each concentration into septa-top vials, each vial was spiked with 31.25 μl of IPA for a final concentration of 50 ppm IPA.

## Preparation of quality control standards

Three Quality Control (QC) standards were prepared at 25 ppm, 60 ppm, and 90 ppm and named QCL, QCM, and QCH, respectively, for low, medium, and high concentrations. Ideally, the QCL should be at the maximum of three times higher than the lower limit of quantitation (LLOQ), which is ten ppm. However, the QC level should not repeat any calibration concentration, so the 25 ppm was determined. They were prepared by directly spiking the urine with the 1/100 ethanol stock. Six QC vials were required, so 32 ml of each QC level was spiked as in Table 5.

The 32-ml QCs were then dispensed into six vials at 5 ml per vial and spiked with 31.25 μl of IPA for a final concentration of 50 ppm IPA. The same QC preparation shall be repeated for each subsequent day of validation. For a full three-day validation, approximately 500 ml of filtered urine is required. The method was validated for its specificity, linearity, intra-day and inter-day inaccuracy and imprecision. The validation method and acceptance criteria were according to Bioanalytical Method Validation, published by the US Food and Drug Administration [31].

## GC-MS parameters and conditions

The GC-MS used was an Agilent 6890 GC system equipped with HP 5973 Mass Selective Detector. Detection was done using a quadrupole detector with an electron impact source. The analytical run was performed on a DB ALC column with 30-m length, 0.32 internal

**Table 4. Serial dilution of calibration standards.**

| Target Concentration (ppm) | Urine Volume (ml) | EtOH Volume (ml) | EtOH Source | Final Volume (ml) |
|---|---|---|---|---|
| 100 | 47.4 | 0.6 | 1/100 Stock | 16.0 |
| 75 | 4.0 | 12.0 | 100 ppm | 16.0 |
| 50 | 20.0 | 20.0 | 100 ppm | 16.2 |
| 30 | 11.6 | 17.4 | 50 ppm | 15.7 |
| 20 | 9.6 | 6.4 | 50 ppm | 16.0 |
| 15 | 8.0 | 8.0 | 30 ppm | 16.0 |
| 10 | 10.6 | 5.3 | 30 ppm | 15.9 |

**Table 5. Preparation details for QC standards.**

| QC Name | Concentration (ppm) | EtOH Volume (ml) | Urine Volume (ml) | Final Volume (ml) |
|---|---|---|---|---|
| QCH | 90 | 0.360 | 31.64 | 32 |
| QCM | 60 | 0.240 | 31.76 | 32 |
| QCL | 25 | 0.100 | 31.90 | 32 |
| LLOQ | 10 | 0.040 | 31.96 | 32 |

diameter, and 1.80 film thickness. Inlet temperature was set at 100˚C. The oven temperature was initially held for two minutes at 40˚C then ramped for 25˚C/min until 250˚C and held for another two minutes, and the total running time was 26 minutes under the scanning mode. Under the selected ion monitoring (SIM) mode, inlet temperature was set at 100˚C and the column temperature at 40˚C without temperature ramping. The column flow was set at 6 mL/min with a total runtime of six minutes. The split mode was used with a split ratio of 25:1. Helium was used as a carrier gas with a flow rate of 0.9 mL/min. Isopropyl alcohol was used as an internal standard (ISTD). A Hamilton 1005SL 5-ml gas-tight syringe was used to withdraw and inject the headspace samples into the GC-MS manually.

Before the injection, the sample was placed in a heat block that had been heated to approximately 80˚C and kept incubating for ten minutes to increase headspace VOCs (S1 Fig). A metal blockade was placed on top of the vial cap to prevent accidental touching of the syringe tip and the liquid. Five milliliters of the headspace sample was injected into GCMS under scanning mode as described previously to determine the compound of interest. After injection, the syringe was thoroughly cleaned with water and dried with an air gun.

### Assessment of sample stability after EVA measurement

Sample evaporation becomes an issue during EVA measurement due to continuous suction of the headspace volume. It was observed that the sample volume got reduced over time. Therefore, the newly developed method was used to quantitate the stability of ethanol content in K12-inoculated urine during measurement.

Due to the destructive nature of the GC-MS assay, multiple spiked samples were prepared to be measured at different time points (S2 Fig). A stock of 25-ppm ethanol-spiked urine stock was prepared and split into nine vials. One vial was quantitated with the GC-MS to confirm the initial concentration. The remaining eight vials were divided into two groups: one group was measured by the EVA, and the other group was not (control). One vial from each group was quantitated by GC-MS after each EVA measurement cycle of ten minutes.

## Results and discussion

### Specificity

The mass over charge number ratios (m/z) 31 and 45 for ethanol and the m/z 45 and 43 for isopropyl alcohol were selected based on pre-scanning of ethanol and IPA. Blank urine sample, blank urine sample spiked with the internal standard, and blank urine sample spiked with 10 ppm EtOH and 50 ppm ISTD were prepared and injected into GC-MS. Two separate peaks were observed for EtOH and ISTD (Fig 3). In particular, ethanol was detected at 4.38 min and IPA at 5.45 min. No significant interfering peaks were found at the retention times at which ethanol and IPA appears. The signal to noise ratios for both drugs was greater than 10. The results demonstrate the adequacy of the method for the specificity of the compounds involved.

### Linearity

A calibration curve was plotted for each validation day for the ratio of EtOH to ISTD response against the concentrations of ethanol (Fig 4). The linearity was observed for all calibration curves performed during method validation with all $R^2$ values above 0.990 (Table 6).

### Within-assay reproducibility

The mean concentration from six replicates of each QC level was calculated, along with their standard deviation and coefficient of variation (Table 7). The mean inaccuracy was calculated

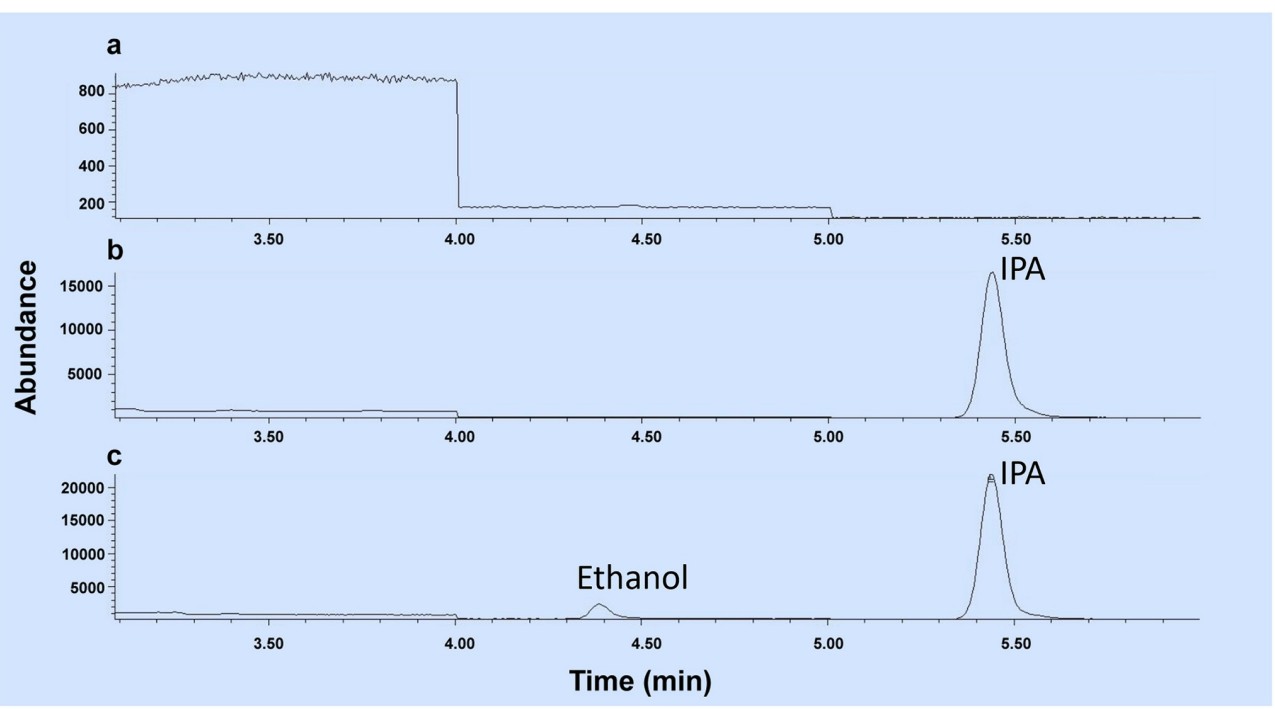

**Fig 3. Chromatograms of (a) blank urine, (b) urine sample spiked with 50 ppm IPA, and (c) urine sample spiked with 10 ppm ethanol and 50 ppm IPA.**

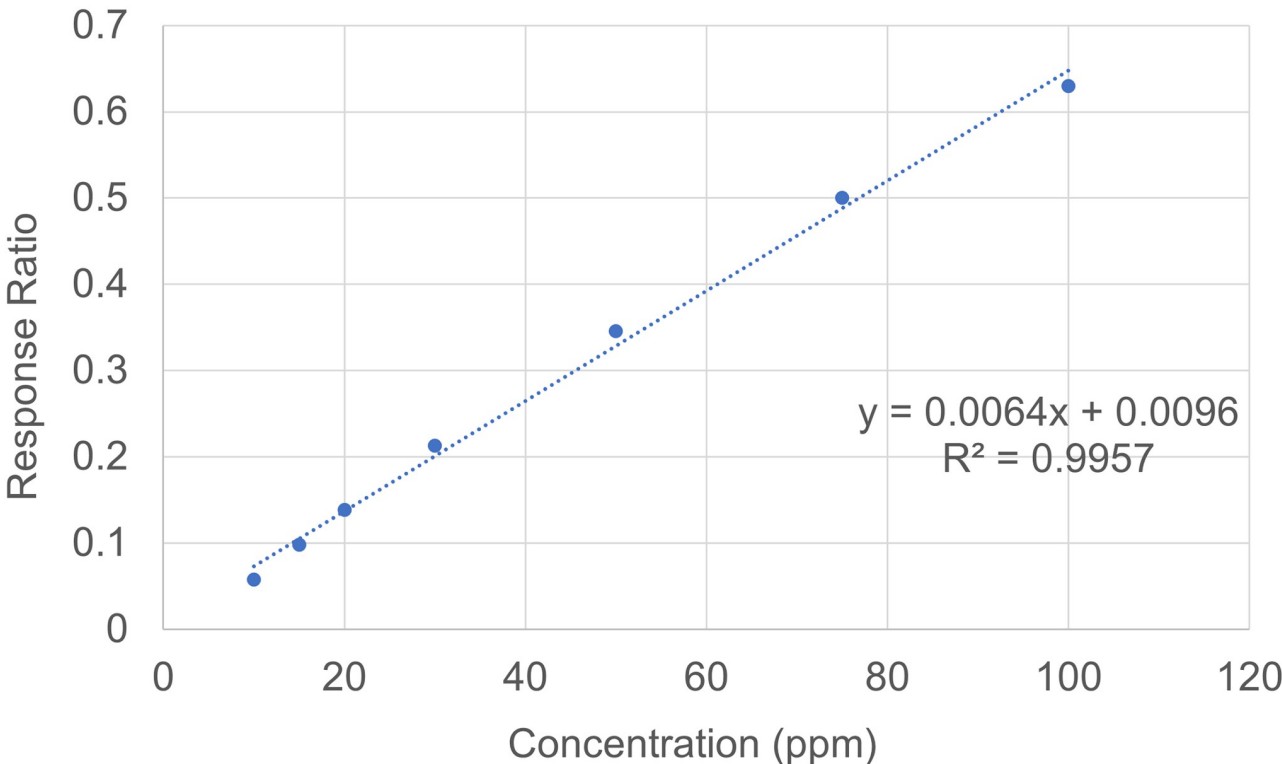

**Fig 4. A representative standard curve.** The y-axis plots the response ratio between the internal standard and the analyte.

**Table 6. Standard curve equations and their coefficients of determination.**

| Validation assay | Linear equation | $R^2$ |
|---|---|---|
| Day 1 | y = 0.0064x + 0.0096 | 0.9957 |
| Day 2 | y = 0.0065x + 0.0068 | 0.9916 |
| Day 3 | y = 0.0063x + 0.0134 | 0.9916 |

**Table 7. Within-assay coefficients of variation and mean inaccuracies.**

| Batch number | QC | Nominal conc. (ppm) | Mean (n = 6) (ppm) | SD (ppm) | CV (%) | Mean inaccuracy (%) |
|---|---|---|---|---|---|---|
| 1 | LLOQ | 10 | 8.50 | 0.83 | 9.76 | 15.00 |
|  | QCL | 25 | 24.24 | 1.31 | 5.40 | 4.9 |
|  | QCM | 60 | 61.59 | 3.19 | 5.18 | 4.74 |
|  | QCH | 90 | 98.01 | 4.77 | 4.87 | 9.01 |
| 2 | LLOQ | 10 | 8.70 | 1.07 | 12.30 | 15.62 |
|  | QCL | 25 | 22.98 | 2.44 | 10.62 | 11.89 |
|  | QCM | 60 | 55.68 | 7.47 | 13.42 | 12.92 |
|  | QCH | 90 | 87.47 | 5.63 | 6.44 | 5.49 |
| 3 | LLOQ | 10 | 11.06 | 0.97 | 8.77 | 10.63 |
|  | QCL | 25 | 25.07 | 3.30 | 13.16 | 8.80 |
|  | QCM | 60 | 62.38 | 3.45 | 5.53 | 4.79 |
|  | QCH | 90 | 89.87 | 4.63 | 5.15 | 4.25 |

by averaging the percent difference between each data point and the mean value. All inaccuracy values were within acceptable ranges according to US FDA guidelines (20% or less for LLOQ and 15% or less for other QCs).

## Between-assay reproducibility

The between-assay repeatability was assessed by calculating the mean, the standard deviation, the CV, and the mean inaccuracy across all 18 samples from the three batches. All mean inaccuracy values were within acceptable ranges according to FDA guidelines (Table 8).

## Quantitation of ethanol in E. coli-inoculated urine

As mentioned in the introduction, there are numerous studies on the e-nose application in detecting and differentiating the causative pathogenic species. Interestingly, most of these studies looked at the VOCs of bacteria in a nutrient agar or jelified urine instead of directly measuring the headspace of the urine sample. Presumably, the solidified form could minimize the evaporation of background components since the presence of water molecules and volatile

**Table 8. Between-assay coefficients of variation and mean inaccuracies.**

| QC | Nominal conc. (ppm) | Mean (n = 18) (ppm) | SD (ppm) | CV (%) | Mean inaccuracy (%) |
|---|---|---|---|---|---|
| LLOQ | 10 | 9.42 | 1.5 | 15.92 | 13.75 |
| QCL | 25 | 24.1 | 2.5 | 10.37 | 8.56 |
| QCM | 60 | 59.88 | 5.69 | 9.50 | 7.48 |
| QCH | 90 | 91.79 | 6.62 | 7.21 | 6.25 |

nutrient broth components could have introduced variability to the sensor system. It is thus important to demonstrate the differentiation capability of an e-nose directly within the head-space of liquid samples for better representation as a point-of-care device. Aathithan et al. analyzed the direct urine samples but did not comment on sample quality, and the reported sensitivity and specificity were not impressive, given that the classification is either positive or negative infection without further strain identification. Here, we devised a benchtop model of UTI by infecting the normal urine samples because it is easy to access, establish, and control in terms of bacterial culture (K12 vs. UPEC) and ethanol concentration. Patient-derived samples could have extremely varied bacterial profiles as well as VOC profiles; that would be beyond the scope of this study. Using a benchtop model, we can decouple confounding factors to understand better the contribution of data quality in discriminating between species. Therefore, the GC-MS method was used to measure samples of K12-inoculated and UPEC-inoculated urine in tandem with an artificial neural network classification of odor data collected by the EVA.

The GC-MS quantitation resulted in 31.33 ppm of ethanol in UPEC-inoculated urine and 18.00 ppm in K12-inoculated urine. Besides the two abovementioned labels, the EVA also measured lab air and normal urine. The dataset was split in half for the training and testing set and evaluated with two-fold cross-validation. The classification resulted in 100% accuracy for both validations, which was determined by taking the ratio between the number of correctly predicted examples over the total number of examples. The VOC fingerprints are visualized by plotting the log of the electrical resistance across all features (Fig 5). The results thus highlighted the potential of EVA to distinguish bacterial strains directly on liquid samples.

These results demonstrated that UPEC and K12 *E. coli* produced distinguishable levels of ethanol, thus causing different VOC fingerprints. The four color-coded labels clearly show different patterns from one another when taking all feature responses as a whole. Without extensive scans, it is not known whether the two strains also have distinct levels of other VOCs. However, the difference in compound concentration enables the differentiation of these two strains of the same species. The GC-MS method allows researchers to directly study the VOC

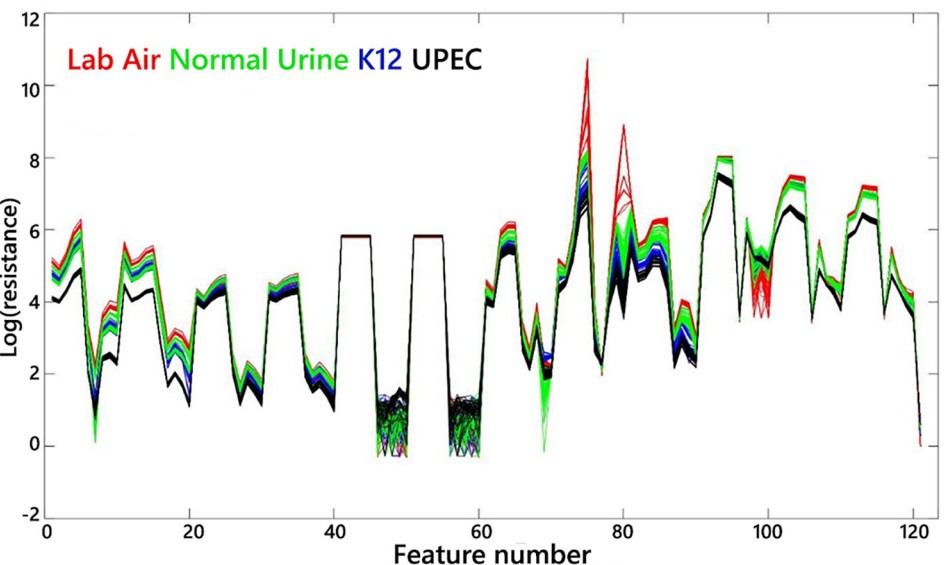

**Fig 5. Visualization of VOC fingerprints shows different patterns in electrical resistance across calculated sample features for each sensor.**

profiles in inoculated urine and aids in decision-making regarding establishing a better measurement protocol and choice of training data for e-nose application.

### Sample instability and consideration for EVA measurement time

The concentration of ethanol-spiked urine was quantitated after every EVA cycle for four cycles. Adhering to the FDA acceptance range, a percent change of 15% or more was deemed significant. Fig 6 shows that the concentration of ethanol diminished significantly (more than 15%) as early as after the third cycle, which is equivalent to 30 minutes of measurement per vial.

Next, a new classification was performed using only three labels of lab air, normal urine, and K12. Data were collected up to 15 cycles each day for two days. The data from the first three cycles were used as a training set to test every subsequent set of three cycles. The results in Table 9 indicate that accurate classification was possible up to nine cycles. Therefore, there are two thresholds for measurement cut-off: the first three cycles and the first nine cycles.

Instability in biological samples, especially urine, has been investigated and shown that long storage time can reduce the emitted VOCs as measured by an e-nose [23]. Typically, the first few examples taken by e-nose are discarded during data processing to avoid variability due to sensor drifting and initial sample instability [32]. However, little is known about sample instability during e-nose measurement. Roine et al. suggested the measurement time can be reduced to 5 minutes based on their classification results [6]. However, this suggestion can only be made after the fact without a priori quantitative basis. Also, a short measurement time gives fewer examples for training the e-nose, a trade-off that needs to be carefully considered. Here, we determined the appropriate measurement time then tested whether using the data

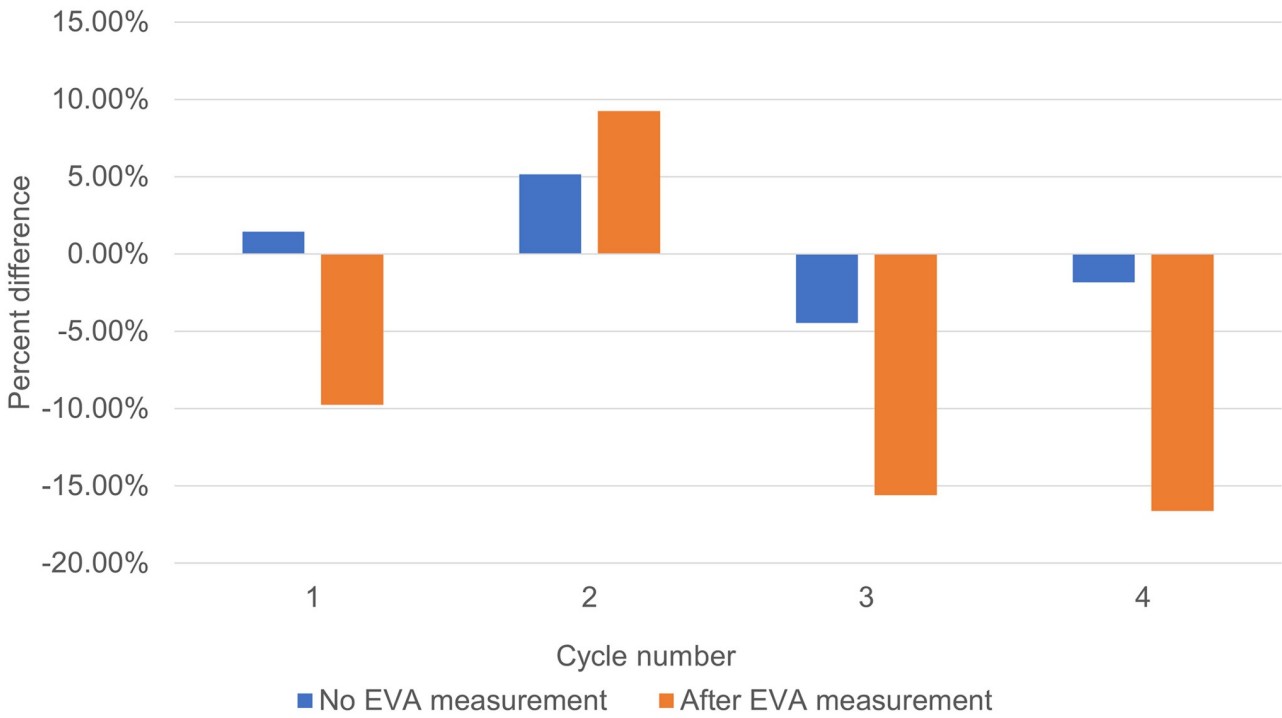

**Fig 6. Percent change in ethanol concentration after each EVA measurement cycle.**

**Table 9. Classification accuracies from training with the first three cycles.**

| Tested on: | Cycles 4–6 | Cycles 7–9 | Cycles 10–12 | Cycles 13–15 |
|---|---|---|---|---|
| Average accuracy: | 95.70% | 98.20% | 55.50% | 37.10% |

within the specified cut-off would yield a better classification than including data after the cut-off.

Between-day classifications were used to evaluate the two cut-offs, and the results are plotted in Fig 7. Training with the first 30 minutes (three cycles) of data gives better accuracies of 97.0% and 73.7% compared to training with the first 90 minutes of data.

Thus, sample instability was induced after 30 minutes of measurement with an e-nose. Specifically, the ethanol concentration was reduced by more than 15%, which is significant according to the FDA guidelines. It was further demonstrated that the ANN model gave superior classification accuracies when using only data from the first 30 minutes of measurement. The result is an example of machine learning basics: data quality is more important than data quantity. The inclusion of data from unstable samples is detrimental to the classification model. The GC-MS method was thus proved useful for determining the measurement cut-off time.

Here, each measurement cycle lasts for ten minutes, which agrees with other studies in the literature [6, 13]. However, there is no standardized protocol for how long this process should be. In a more recent study by Capelli et al., the urine headspace was flown into the sensor chamber for 50 minutes [33]. It is thus possible to rerun sample collection to increase the

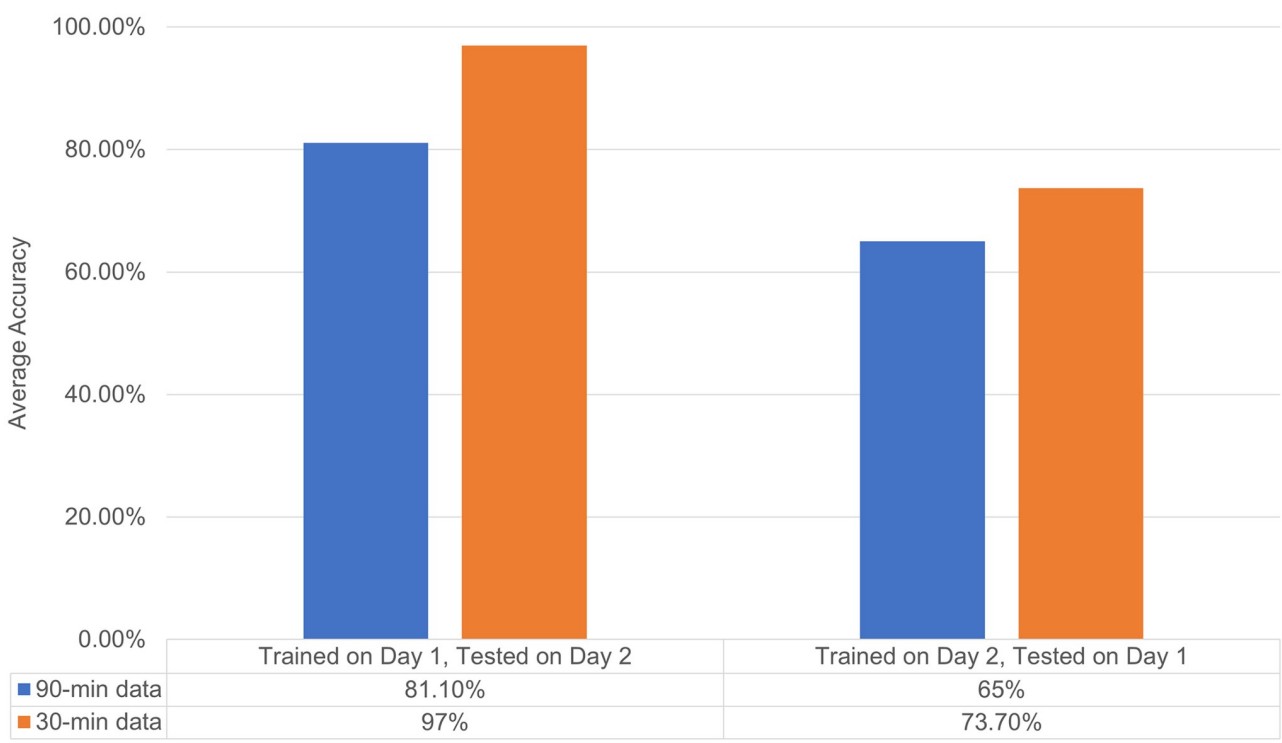

| | Trained on Day 1, Tested on Day 2 | Trained on Day 2, Tested on Day 1 |
|---|---|---|
| ■ 90-min data | 81.10% | 65% |
| ■ 30-min data | 97% | 73.70% |

**Fig 7. Cross-validation accuracies between days for 90-min and 30-min measurements.**

number of training examples for the model as long as the cut-off time can be determined, which could improve the efficiency of this assay.

Since e-noses typically output electrical resistances, which result from the collective interaction of the whole VOC profile with the sensors, they lack the credibility to aid in decision-making regarding quality control of the data. Furthermore, we recognize that the VOC profiles in real patient-derived UTI samples are likely more complicated. For example, *E. coli* is not the only species that produce ethanol as a by-product, but other UTI-causing strains such as *Klebsiella aerogenes*, albeit less prevalent, also ferment lactose and release ethanol. It would be beneficial to strengthen the sensitivity and specificity of the sensors toward ethanol. However, the EVA will be designed to distinguish various bacterial strains, many of which may produce compounds other than ethanol. Thus, we must also include a variety of sensors to account for this dynamicity. These sensors are modular and can be added, removed, or replaced. Therefore, we could optimize a sensor array that targets the most prevalent strains associated with UTI.

There are a number of limitations to this study that should be acknowledged. The most notable one is that we tested the EVA on inoculated urine in developing our assay. While providing consistency in sample preparation in a proof-of-concept setting, we cannot project the results for clinical samples in which urine can vastly vary in texture and contents, causing a lot of noise for the neural network. It is important to analyze the differential concentration of VOCs in healthy vs. UTI to screen for appropriate sensors that respond to the top compounds with the most difference. A second limitation is the drifting of sensors. We did not specifically assess the age of the sensors used in our experiment. Bax et al. showed that classification using one-year-old sensors was much worse than using new ones, and they proposed a correction model that significantly improved the performance from 55% to 80% [34]. Regarding drift, baseline shift among analyses performed on different days could also be seen as a shortcoming of the study, as ambient temperature or humidity could influence baseline sensor readings. We did not observe a significant shift in baseline over the experimental duration. However, Bax et al. also described a pre-treatment procedure to compensate for baseline drift by using Standard Normal Variate. The same procedure can be adapted to improve our model. A third limitation is that the testing time of 10 minutes for each sampling cycle could have been a little too long. As aforementioned, a shorter test time has been proposed [6]. We could further reduce the length of a single measurement from 80 seconds.

Future studies should carefully evaluate the degree of sample degradation. While sample instability is inevitable, it is efficient to determine the extent of this degradation to maximize data quantity without a trade-off for data quality. The same headspace analysis should be considered in every future experimental design to determine the ground truth, which is the concentration of analytes, before proceeding to use the collected data. By characterizing the sample stability during measurement, one can safely reduce variability in the data due to the very act of measurement itself while being able to incorporate background compounds as part of the VOC profile.

## Conclusion

In this paper, we demonstrated the validation and quantitation data of ethanol in the headspace of urine samples inoculated with *E. coli*. The method validation fulfilled all the criteria as outlined in the Bioanalytical Method Validation protocol published by the US FDA. The method was successfully applied on ethanol measurement in samples of K12 and UPEC-inoculated urine as an initial step towards improving the outcome of VOC measurement by electronic nose technology through a validated headspace GC-MS method. The main interest of

using this method was to characterize *E. coli*-inoculated urine samples that are prepared for training an electronic nose to detect urinary tract infection and differentiate between different causative agents. By using the new method, it was shown that different strains of *E. coli* could produce different levels of ethanol concentration, making it possible to differentiate between them based on distinct VOC fingerprints. The quantitation also revealed that e-nose measurement could affect sample stability over time. Therefore, the cut-off time for future measurement should be wisely determined to avoid the collection of unusable data due to a reduction in compound concentration. The GC-MS will continue to be a useful tool to support technology development, in characterizing samples to build a useful training library for UTI detection directly through liquid samples without extra preparation steps; thus enabling next generation real-time and point-of-care diagnosis of UTI.

## Supporting information

**S1 Fig. Headspace sample being withdrawn from urine vial heated in a heat block.** The real temperature was measured in an adjacent water vial by a digital thermometer. Inset: Urine sample in a septa-top vial for headspace analysis.
(TIF)

**S2 Fig. A schematic of urine sample stability test with GC-MS.**
(TIF)

**S3 Fig. Sample set up with the EVA.**
(TIF)

## Author Contributions

**Conceptualization:** Nam Than, Zamri Chik, Luisa Bozano, Aminat Adebiyi.

**Data curation:** Nam Than, Zamri Chik, Amy Bowers, Aminat Adebiyi.

**Formal analysis:** Nam Than.

**Methodology:** Nam Than, Zamri Chik, Amy Bowers, Luisa Bozano, Aminat Adebiyi.

**Software:** Aminat Adebiyi.

**Supervision:** Aminat Adebiyi.

**Writing – original draft:** Nam Than, Zamri Chik.

**Writing – review & editing:** Nam Than, Zamri Chik, Amy Bowers, Luisa Bozano, Aminat Adebiyi.

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
