## [Decision Letter · Decision Letter 0]

6 Dec 2021

PONE-D-21-22287Quantitation of ethanol in UTI assay for volatile organic compound detection by electronic nose using the validated headspace GC-MS methodPLOS ONE

Dear Dr. Adebiyi,

Thank you for submitting your manuscript to PLOS ONE. After careful consideration, we feel that it has merit but does not fully meet PLOS ONE’s publication criteria as it currently stands. Therefore, we invite you to submit a revised version of the manuscript that addresses the points raised during the review process.

ACADEMIC EDITOR: As appended below, the reviewers have raised major concerns/critiques (reviewer # 3 is against publication) and suggested further justification/work to consolidate the findings. Do go through the comments and amend the MS accordingly.

We look forward to receiving your revised manuscript.

Kind regards,

A. M. Abd El-Aty

Academic Editor

PLOS ONE

Journal Requirements:

2. Please include in your Data availability statement details of how other researchers may access the prototype information and any equipment or data used in this study which is not public. Please include information on how another researcher would enter into a a Joint Study Agreement with IBM Research, plus a non-author IBM contact for facilitating that process. Please also clarify whether the authors had any special privileges in accessing proprietary equipment or data."

3. Please complete the Financial disclosure statement and Competing interests statement, including all financial and materials support, as well as any author commercial affiliations."

4. Please include a caption for figure 8.

5. Please ensure that you refer to Figure 7 in your text as, if accepted, production will need this reference to link the reader to the figure

Reviewers' comments:

Reviewer's Responses to Questions

**Comments to the Author**

1. Is the manuscript technically sound, and do the data support the conclusions?

Reviewer #1: Yes

Reviewer #2: Partly

Reviewer #3: No

Reviewer #4: Yes

2. Has the statistical analysis been performed appropriately and rigorously? 

Reviewer #1: Yes

Reviewer #2: No

Reviewer #3: Yes

Reviewer #4: Yes

3. Have the authors made all data underlying the findings in their manuscript fully available?

Reviewer #1: No

Reviewer #2: No

Reviewer #3: Yes

Reviewer #4: Yes

4. Is the manuscript presented in an intelligible fashion and written in standard English?

Reviewer #1: Yes

Reviewer #2: No

Reviewer #3: Yes

Reviewer #4: Yes

5. Review Comments to the Author

Reviewer #1: “Quantitation of ethanol in UTI assay for volatile organic compound detection by

electronic nose using the validated headspace GC-MS method” is an excellent study where the authors used inoculated urine samples as a proof-of-principle to show the ability of the EVA IBM e-nose to distinguish between two E. coli species. Interestingly, the authors explored the relationship between volatile organic compound (VOC) - ethanol - and e-nose result accuracy. The neural networks of e-noses do not offer an insight into the composition of the headspace. In contrast, GC-MS can shed light on the VOCs present – using this tool the authors show that just testing the sample with the e-nose degrades the quality of the sample over time. Thus, one of the most important takeaways from this study is that the quality of the data is more important than the quantity of data. Furthermore, this study advances the field, paving the way to faster and cheaper UTI diagnosis using e-noses by showing the abilities of the EVA to accurately discriminate between samples. Simultaneously, it builds up a GC-MS method to study headspace VOC with the view of gathering data for better choice and calibration of the sensors in the e-nose.

This study follows up on a conference paper in 2019: “Rapid Strain Differentiation of E. coli-inoculated Urine Using Olfactory-based Smart Sensors”. It was interesting to see the progress done since then. The manuscript submitted by the authors is the only use of EVA IBM e-nose on urine samples I could find in the literature.

Minor

There are some points that need to be addressed:

1. The calculations in table 4 are correct based on the formula provided in line 236 (the whole paragraph, line 234-239, is very useful – well done for including!). However, the final volume column is confusing as sometimes the urine volume + EtOH volume does not add up to the final volume. For example, target concentration 100 ppm 47.4 (Urine volume) + 0.6 (EtOH volume) is 48 ml not 16 ml (Final volume written in the table). Similarly, this happens again for target concentration 50 ppm and 30 ppm.

The other target concentrations do add up to what is in the table. I think this needs to be clarified.

2. Table 5 (line 266) appears to have an error: the EtOH volume appears to be wrong – when used for calculation it does not give the target concentration or the final volume (EtOH volume 0.008 ml + urine volume 31.96 volume = 31.968 ml total volume; not 32 ml as written in the table). The correct concentration in ppm, as well as the final volume, works out with 0.04 ml of EtOH.

3. Line 94 “Table 1 is limited to those used in urinary pathogen detection” and line 357 states that e-nose studies previously looked at nutrient agar or gelified urine instead of direct urine measurements.

There have been previous e-nose studies that use urine directly or looked at urinary pathogen detection such as: Aathithan, S.; Plant, J.C.; Chaudry, A.N.; French, G.L. Diagnosis of Bacteriuria by Detection of Volatile Organic Compounds in Urine Using an Automated Headspace Analyzer with Multiple Conducting Polymer Sensors. J. Clin. Microbiol. 2001, 39, 2590–2593.

A review that summarises the literature is available: Dospinescu, V.-M., A. Tiele & J. A. Covington (2020) Sniffing out urinary tract infection—Diagnosis based on volatile organic compounds and smell profile. Biosensors, 10, 83.

Table 3 in the aforementioned review encapsulates the studies published in the field and provides an ampler overview of the experiments previously conducted on VOCs and infected urine/pathogens involved in UTI.

4. Presentation/Grammar:

The paragraph between lines 240-246 is repeated (lines 247-253).

Line 284 - “temperature was set at 100oC” should be 100°C.

Line 71 – “urines” should be urine.

Once again, I believe this study advances the field and shows that the IBM EVA e-nose has the potential to be used as a diagnostic tool for UTI with enough developments. In addition, it also provides a framework that could be used for other VOCs (not just ethanol) for optimising e-nose use based on GC-MS data. My recommendation is that the study should be published once the points above are addressed.

Other comments:

1- The text is sufficiently detailed to understand the work whilst also being succinct. It reads easily and keeps the reader engaged. The problem is framed in the introduction and the aim is identified in lines 105-107.

2- E. Coli ferments lactose (or arabinose) and as a by-product it produces ethanol. However, this is not specific to E. coli, another agent that causes UTI (although not as common) also produces ethanol: Klebsiella aerogenes.

3- Data availability:

The authors state “Yes - all data are fully available without restriction” but in the next section “Data access are available upon request” together with two links (“GC-MS data are available at

https://app.box.com/folder/125090388794 and Electronic nose EVA data at

https://app.box.com/folder/113625571232”).

However, even after signing up on the website, accessing the link gives the following error “Oops! We can't seem to find the page you're looking for.” - Considering how "The PLOS Data policy requires authors to make all data underlying the findings described in their manuscript fully available without restriction" this app.box issue should be addressed.

Reviewer #2: The paper reports on the use of an electronic nose for measuring ethanol concentration in urine samples. It is really about using validation methods on artificial urine samples and I think the title is a little miss leading as no samples from UTI sufferers are included. A large percentage of the paper is given over to GCMS and the application of a standard to these measurements. The authors should be clear that this is really the purpose of the paper and change the focus on the paper. I then feel it would be more interesting to the community. The introduction needs a fair amount of work and some of the statistics need checking. Also, some of the purposes of the tests need to be explained in context to UTI. For example, a 30 minute electronic nose measurement is a very long time. I would like to see more common electronic nose analysis – so a simple PCA of the features would be useful. In terms of sample re-runs, this is not something that you would undertake with an electronic nose and there are a number of papers that discuss sample testing optimisation (which are not referenced here). The paper needs to be re-structured and re-written with a different focus.

Some specific comments are below.

Line 50: It is a little odd to start a urine based study mentioning breath. Maybe this could just list the biological sources?

Line 54: Poor English. Also no details are provided about the study. Was it an infection or kidney failure or?

Line 59: Was this of just the bacteria or an infected human sample?

Line 62: Don’t understand what you are trying to say.

Line 65: I disagree with this statement. A 10 cent dipstick test will inform you if you have UTI. More important would be to identify the bacterial species or identify UTI in those where there is a high false positive rate.

Line 69: I disagree with this statement. How can static headspace analysis be better that a trap pre-concentration step? Or do you mean something else? Also, I am not convinced this paragraph adds anything. I don’t think you need to prove that headspace analysis is not a valid approach.

Line 83: You should give a better explanation (or more classical explanation) of what an electronic nose is and then remove table 1. I would give examples in the introduction of previous UTI student and then compare your result with the literature in a table at the end of the work. I am unsure what the relevance of a prostate cancer study, but you could mention earlier the relevance of cancer urine studies (bladder, prostate, colorectal etc.).

I don’t think table 2 is relevant. I would like to see some focus on previous literature showing that ethanol is important (which isn’t discussed) and how it is modulated in the presence of disease. Also, some comment on the biological pathway that creates ethanol – from host response or from the bacteria itself. This should be included in the introduction.

Line 144: You are working on detecting ethanol, but many of the sensors are not targeting ethanol, why is this? Also, it is an odd choice of sensors. Was there a reason for this combination from different manufacturers? Was any optimisation of the array undertaken?

Line 158: Please provide technical details of how the unit was driven and measured. For example, you provide heater voltage as a %, not as a V. Where all the sensors operated with the same temperature pulses or were they different?

Line 163: How was this optimisation undertaken?

Line 168: How was resistance measured with a fixed voltage? What was the internal volume of the chamber? Was there a background reading before the measurement or were you just using the temperature modulation to give you an non-sensing resistance? Or was something else done?

Line 216: Please provide details of the vials and how they were modified/used to allow an air intake. What was the tubing used?

Line 223: Not clear how you extracted features from the raw data.

Line 227: What software/program/model was used to create the BNN?

Line 229: Were the samples from the same sample excluded from the training set for when they were being used as a test set? Otherwise, you are training and testing on the same samples. It is not clear how you are doing the cross-validation.

Line 240: How was this done?

Line 271: Was this human urine or artificial?

Line 323: Figure 3 is really difficult to read and there doesn’t appear to be any axes labels on the figure (though it might be the poor quality of the image). I wonder if the PDF process has caused this?

Line 358: I am pretty sure there are some UTI studies using direct analysis. The authors should comment on these papers as well.

Line 397: Why is this important? Would not the sample be tested and then disposed of? If a second reading was needed, they would just take some more out of the sample container – or just get more urine from patient. The reason for doing this needs to be explained. You are providing evidence that you should just test once.

Line 415: Why would you measure for 30 minutes with an electronic nose? For what purpose?

Line 427: There have been a number of studies looking at urine stability with electronic noses – which I noticed are not referenced. Also, the result found here is well known in the electronic nose community.

Line 437: This is really important – and much more that the focus on electronic noses. I would like this to have been in the introduction.

Line 445: Would this not be dependent on the level of infection in real life?

Reviewer #3: The capabilities of quantifying individual VOCs is not normally an important requirement for identification of diseases or pathogens responsible for causing diseases when using electronic-nose devices. The most important information to validate is the identities of VOCs making up the E-nose smellprint signatures and thus VOC profiles, not concentration of VOCs which is normally only needed in metabolomic studies to determine effects of pathogens on metabolic pathways of the host. Thus, quantification does not add significantly to the capabilities for testing the efficacy of a new experimental e-nose device. The ultimate objective was to develop the capabilities of the e-nose for detecting UTI caused by different microbes. The methods developed here do not contribute to that objective and the data obtained is normally part of a pilot study for methods development and not published as a stand alone unit without e-nose data on UTI samples from different types of microbial causes with adequate controls. Quantification of a single possible VOC in a UTI sample headspace provides very little information towards development of an e-nose library database containing specific complex mixtures of VOCs that affect the output signatures of the e-nose sensor array. The sensor array responds to all of the VOCs present in the headspace, not just a single VOC such as ethanol.

All of the figures (Fig. 1-7) are of very low quality resolution and do not provide the data necessary to support the efficacy of a new experimental e-nose device for UTI diagnostics (based on quantitation of EtOH alone). Development of methods for quantitation of EtOH along with a standard curve alone do not contribute significantly towards development of e-nose methods useful for UTI diagnostics and thus the Conclusions are not supported by the objectives of the study or the data obtained towards this purpose.

Reviewer #4: The paper describes the use of an electronic nose to distinguish the pathogens causing urinary tract infection, and proposes an new experimental method to improve sample stability during e-nose measurements. The problem of sample stability during e-nose measurment should be better claimed in the introduction, because it is the focus of the described sperimentation.

The methods involved to prepare EVA samples and GC-MS calibrants are described properly. However, the authors should clearly state the reasons leading to the use of normal urine samples to be inoculated rather than real infected samples. I think that authors should also better describe the approach involved for training the e-nose. Specifically, they should describe the scheme involved for presenting samples to the e-nose and validating the classification performance. The paragraph (lines 223-231) is not clear. The value of the sensor resistance at different times was used as features?

All figures of the paper have a very poor quality. They should be revised. In some cases, the text is not readable.

Moreover, the authors should add reference supporting their statments throughout the paper.

6. PLOS authors have the option to publish the peer review history of their article (what does this mean?). If published, this will include your full peer review and any attached files.

Reviewer #1: No

Reviewer #2: No

Reviewer #3: No

Reviewer #4: **Yes: **Carmen Bax

---

## [Author Response · Author response to Decision Letter 0]

28 Apr 2022

RESPONSE TO REVIEWERS (PONE-D-21-22287)

We appreciate the reviewers’ thoughtful comments and insight. We are pleased to receive many positive comments. However, we acknowledge several concerns raised by the reviewers. We have addressed these shortcomings in the revised manuscript. Response to reviewers is detailed as follows. We look forward to hearing from you regarding our revision. We would be glad to respond to any further questions and comments.

Reviewer 1:

1. "Quantitation of ethanol in UTI assay for volatile organic compound detection by electronic nose using the validated headspace GC-MS method" is an excellent study where the authors used inoculated urine samples as a proof-of-principle to show the ability of the EVA IBM e-nose to distinguish between two E. coli species. Interestingly, the authors explored the relationship between volatile organic compound (VOC) - ethanol - and e-nose result accuracy. The neural networks of e-noses do not offer an insight into the composition of the headspace. In contrast, GC-MS can shed light on the VOCs present – using this tool the authors show that just testing the sample with the e-nose degrades the quality of the sample over time. Thus, one of the most important takeaways from this study is that the quality of the data is more important than the quantity of data. Furthermore, this study advances the field, paving the way to faster and cheaper UTI diagnosis using e-noses by showing the abilities of the EVA to accurately discriminate between samples. Simultaneously, it builds up a GC-MS method to study headspace VOC with the view of gathering data for better choice and calibration of the sensors in the e-nose. This study follows up on a conference paper in 2019: "Rapid Strain Differentiation of E. coli-inoculated Urine Using Olfactory-based Smart Sensors". It was interesting to see the progress done since then. The manuscript submitted by the authors is the only use of EVA IBM e-nose on urine samples I could find in the literature.

We appreciate the positive comments from Reviewer 1 on the merit of this study and the promising perspectives of our EVA e-nose. Indeed, the key takeaway in this article is that the quality of the data could be a function of the data collection method and thus should be carefully controlled.

Minor comments

2. The calculations in table 4 are correct based on the formula provided in line 236 (the whole paragraph, line 234-239, is very useful – well done for including!). However, the final volume column is confusing as sometimes the urine volume + EtOH volume does not add up to the final volume. For example, target concentration 100 ppm 47.4 (Urine volume) + 0.6 (EtOH volume) is 48 ml not 16 ml (Final volume written in the table). Similarly, this happens again for target concentration 50 ppm and 30 ppm. The other target concentrations do add up to what is in the table. I think this needs to be clarified.

The final volume column reflects the ultimate volume after transferring some amount to make lower dilutions, not the initially generated amount. For example, at 100 ppm, 48 ml is initially generated, but a total of 32 ml (3rd column) is used to make the target 75-ppm and 50-ppm concentrations, hence the remaining 16 ml. We added a sentence to clarify this point.

3. Table 5 (line 266) appears to have an error: the EtOH volume appears to be wrong – when used for calculation it does not give the target concentration or the final volume (EtOH volume 0.008 ml + urine volume 31.96 volume = 31.968 ml total volume; not 32 ml as written in the table). The correct concentration in ppm, as well as the final volume, works out with 0.04 ml of EtOH.

We thank Reviewer 1 for pointing out this error. It has been fixed.

4. Line 94 "Table 1 is limited to those used in urinary pathogen detection" and line 357 states that e-nose studies previously looked at nutrient agar or gelified urine instead of direct urine measurements. There have been previous e-nose studies that use urine directly or looked at urinary pathogen detection such as: Aathithan, S.; Plant, J.C.; Chaudry, A.N.; French, G.L. Diagnosis of Bacteriuria by Detection of Volatile Organic Compounds in Urine Using an Automated Headspace Analyzer with Multiple Conducting Polymer Sensors. J. Clin. Microbiol. 2001, 39, 2590–2593.

We appreciate the reference suggested by Reviewer 1. Aathithan et al. using artificial urine inoculated with urinary pathogens and clinical urine samples is a great approach, making it a helpful reference. However, they used PCA with only two labels (infection positive or negative) which is not as powerful as the artificial neural network that we used. This point was reflected in the revised manuscript (lines 97-103).

A review that summarises the literature is available: Dospinescu, V.-M., A. Tiele & J. A. Covington (2020) Sniffing out urinary tract infection—Diagnosis based on volatile organic compounds and smell profile. Biosensors, 10, 83. Table 3 in the aforementioned review encapsulates the studies published in the field and provides an ampler overview of the experiments previously conducted on VOCs and infected urine/pathogens involved in UTI.

We are thankful for the very detailed table summarizing past studies on VOCs. We reflect this review in the introduction to highlight the potential of e-nose research (lines 65) 

5. Presentation/Grammar:

The paragraph between lines 240-246 is repeated (lines 247-253).

Line 284 - "temperature was set at 100oC" should be 100°C.

Line 71 – "urines" should be urine.

We have fixed these mistakes.

Other comments:

6- E. Coli ferments lactose (or arabinose) and as a by-product it produces ethanol. However, this is not specific to E. coli, another agent that causes UTI (although not as common) also produces ethanol: Klebsiella aerogenes.

We appreciate this useful comment. Indeed, future studies can look more into the classification between K. aerogenes and E. coli and expand the profiles of VOCs measured by GC-MS to better design an e-nose capable of distinguishing them. We reflected this point in the Result and Discussion section (line 456)

7- Data availability:

The authors state "Yes - all data are fully available without restriction" but in the next section "Data access are available upon request" together with two links ("GC-MS data are available at

https://app.box.com/folder/125090388794 and Electronic nose EVA data at

https://app.box.com/folder/113625571232").

However, even after signing up on the website, accessing the link gives the following error "Oops! We can't seem to find the page you're looking for." - Considering how "The PLOS Data policy requires authors to make all data underlying the findings described in their manuscript fully available without restriction" this app.box issue should be addressed.

We have uploaded the data files to a public repository (Kaggle.com):

- GCMS data – raw files used to develop validation table - https://www.kaggle.com/datasets/aminatadebiyi/beva-plos-one

- IBM EVA data and features for Artificial Neural Network - https://www.kaggle.com/datasets/aminatadebiyi/beva-plos-one-ann

Reviewer 2

1. The paper reports on the use of an electronic nose for measuring ethanol concentration in urine samples. It is really about using validation methods on artificial urine samples and I think the title is a little miss leading as no samples from UTI sufferers are included. 

We appreciate the valuable comment. However, we used normal human urine inoculated with E. coli to create a simple model of UTI, hence “UTI assay” in the title. The purpose of the paper is not to measure ethanol concentration but to show how sample quality can be affected by the very act of e-nose measurement and a proof of concept for distinguishing different strains of E. coli in inoculated urine samples. 

2. A large percentage of the paper is given over to GCMS and the application of a standard to these measurements. The authors should be clear that this is really the purpose of the paper and change the focus on the paper. I then feel it would be more interesting to the community. The introduction needs a fair amount of work and some of the statistics need checking. 

We clarify that the purpose of this study is to demonstrate the e-nose ability to distinguish K12 and UPEC E. coli in inoculated urine and that the classification could be further improved if sample quality can be controlled by the quantification of a characteristic VOC (ethanol). The GC-MS method development, therefore, is secondary to the main purpose of showing a successful classification.

We agree that the GC-MS method development takes a large portion of the paper, which is necessary because a specific method for the quantification of headspace ethanol in urine is uncommon. We switched a paragraph (lines 141) and revised the text regarding the purpose of the study to clarify the focus of the paper.

3. I would like to see more common electronic nose analysis – so a simple PCA of the features would be useful. 

The reviewer is encouraged to check our previous report on the PCA and other classifiers. Adebiyi, A. et al. Sensors & Transducers Using Olfactory-based Smart Sensors. 238, 94–98 (2019). In this paper, we focused on demonstrating as a proof-of-concept the capability of an ANN model instead of comparing between multiple model, but we found the combination of our Feature Tuning approach with the ANN to maintain the robustness of our model over a longer period, probably due to sensor drift. Details of our feature tuning approach can be found here https://patents.google.com/patent/US20210172919A1/en. 

4. Also, some of the purposes of the tests need to be explained in context to UTI. For example, a 30-minute electronic nose measurement is a very long time. In terms of sample re-runs, this is not something that you would undertake with an electronic nose and there are a number of papers that discuss sample testing optimisation (which are not referenced here). 

We understand the concern. To clarify, each measurement lasts 10 minutes, yielding a total of seven training/testing examples (80 s per example) for the artificial neural network model. This time length agrees well with previous studies (e.g., Asimakopoulos et al. 2014, and Roine et al. 2014). A recent study even flowed the headspace into their eNose for 50 minutes (Capelli, L., Bax, C., Grizzi, F. et al. Optimization of training and measurement protocol for eNose analysis of urine headspace aimed at prostate cancer diagnosis. Sci Rep 11, 20898 (2021)). We updated the manuscript to reflect these references (line 247).

The 10-min measurement cycle was repeated several times to determine the optimal cut-off. By doing so, we achieve two goals. First, there was more frequent flushing in between to purge the sensor chamber. Second, we can maximize the number of training examples instead of discarding the sample after only 10 minutes of measurement. We show that we can include data up to 30 minutes (3 measurement cycles) before the sample degraded beyond the acceptable 15% fluctuation.

Specific comments

Line 50: It is a little odd to start a urine based study mentioning breath. Maybe this could just list the biological sources? Line 54: Poor English. Also no details are provided about the study. Was it an infection or kidney failure or?

We revised these lines to clear up unnecessary information.

Line 59: Was this of just the bacteria or an infected human sample?

It is of the bacteria themselves, according to a review by Bos et al., as mentioned in the sentence following that line.

Line 62: Don’t understand what you are trying to say.

We revised this sentence to clarify that even though multiple species can commonly produce a VOC, the headspace concentration could be different, thus providing the basis for bacterial discrimination. 

Line 65: I disagree with this statement. A 10 cent dipstick test will inform you if you have UTI. More important would be to identify the bacterial species or identify UTI in those where there is a high false positive rate.

We agree with the reviewer that a high false-positive rate could contribute to the overuse of broad-spectrum antibiotics, in which case identification of bacterial species would be critical. However, the gold standard for UTI detection is bacterial culture which takes time. In many cases, patients can be given broad-spectrum antibiotics as a precautionary measure. A rapid test can thus lower this likelihood. A dipstick is also a rapid test, but it does not have specific identification like the e-nose. We reflected this point in the revised version.

Line 69: I disagree with this statement. How can static headspace analysis be better that a trap pre-concentration step? Or do you mean something else? Also, I am not convinced this paragraph adds anything. I don’t think you need to prove that headspace analysis is not a valid approach.

We are not proving that headspace analysis is not a valid approach or that it is better than a trap analysis. In this paragraph, we reason our choice for the static headspace method as it is relatively simple, has similar sensitivity to trap analysis or dynamic purge, and avoids time- and cost-consuming steps.

Line 83: You should give a better explanation (or more classical explanation) of what an electronic nose is and then remove table 1. I would give examples in the introduction of previous UTI student and then compare your result with the literature in a table at the end of the work. I am unsure what the relevance of a prostate cancer study, but you could mention earlier the relevance of cancer urine studies (bladder, prostate, colorectal etc.).

We appreciate the thoughtful suggestion. However, we think Table 1 is necessary to highlight sensor type (MOX) and algorithm (ANN) choice in our study. Since our scope is to highlight the GC-MS validation of our approach, we are happy to take out the table if it is confusing.

I don’t think table 2 is relevant. I would like to see some focus on previous literature showing that ethanol is important (which isn’t discussed) and how it is modulated in the presence of disease. Also, some comment on the biological pathway that creates ethanol – from host response or from the bacteria itself. This should be included in the introduction.

We added a reference for a review paper in the introduction (Dospinescu et al. 2019). This paper excellently reviewed the VOC profiles on relevant UTI-causing species. We commented on the ethanol production by E. coli in the introduction based on this paper.

Line 144: You are working on detecting ethanol, but many of the sensors are not targeting ethanol, why is this? Also, it is an odd choice of sensors. Was there a reason for this combination from different manufacturers? Was any optimisation of the array undertaken?

The IBM Electronic Volatile Analyzer™ is designed to be modular and is comprised of a variety of diverse sensors to respond to a variety of analytes, therefore to achieve orthogonality of sensors, we included the selection described. This specific combination of sensors was selected using an analytic-driven approached outlined in our patent “Sensor tuning - sensor specific selection for IoT-electronic nose application using gradient boosting decision trees.” https://patents.google.com/patent/US20210172918A1/en This combination of sensors and features provided the most robust classification for this biological application, which is why it was selected. 

Line 158: Please provide technical details of how the unit was driven and measured. For example, you provide heater voltage as a %, not as a V. Where all the sensors operated with the same temperature pulses or were they different?

The design of the platform is modular: each gas sensor is mounted on its own sensor module, a Printed Circuit Board (PCB) of common design carrying an integrated microcontroller with built-in heating components, and the circuitry needed to operate the sensor. The sensor modules communicate via I2C protocol with a central hub (BeagleBone Black), a single-board computer that orchestrates the sensor modules and processes the multisensorial output. 

Each MOX sensor was operated using periodic waveform of heater voltage, expressed as a percentage of the maximum operating voltage Vmax recommended by each sensor manufacturer. This resulted in a stepwise modulation of the device temperature (Figs. 2b(i)&(ii)), a well-known technique that can be used to enhance and tune the dependence of the sensing element resistance to the surrounding environment. 

Although the duration and amplitude of the individual waveforms was adjusted independently for each sensor, all waveforms were synchronized to a period of 80 s to simplify the handling and processing of the sensor array outputs. 

Line 163: How was this optimisation undertaken?

More details on our optimization techniques can be found at our patent filing, “Adaptive sensor temperature control for fast recovery,” highlighting our temperature control optimization method to maximize sensor response to our target analyte. (https://patents.google.com/patent/US20210063372A1/en)

Line 168: How was resistance measured with a fixed voltage? What was the internal volume of the chamber? Was there a background reading before the measurement or were you just using the temperature modulation to give you an non-sensing resistance? Or was something else done?

The data was collected at each given voltage at 10Hz, but the full sensor response was characterized as the time period of the operational voltage sequence (80s), that consisted of multiple voltage steps, giving us the response and recovery pattern of the sensor’s exposure to the sample analyte. 

Rather than the use of background readings to minimize sensor drift, we performed system stabilization for an hour before each experiment. This involved exposing the sensors for an hour to lab air in ambient conditions. This stabilization method in combination with our choice of feed-forward neural network, captured the variation in our readings to maintain a robust system for the given measurements.

The interal volume of the chamber was about ~700ml. 

Line 216: Please provide details of the vials and how they were modified/used to allow an air intake. What was the tubing used?

The vials used are Wheaton Septa-cap Vials. They have a rubber top cap that can be punctured with a needle. The tubing used was made from silicon. We provided an additional supplementary figure (S3) to show our set up.

Line 223: Not clear how you extracted features from the raw data.

Our feature extraction was achieved through an amplitude-driven approach of extracting the mean area under the curve for the given response duration. To focus the scope of the paper on the GC-MS validation, we did not go into much detail on this approach, however more details on this approach and the reasoning for its application can be found in our patent-filing, “Feature tuning – application dependent feature type selection for improved classification accuracy.” https://patents.google.com/patent/US20210172919A1/en

Line 227: What software/program/model was used to create the BNN?

GNU Octave v-5.1.0.0 (GUI), available through open source.

Line 229: Were the samples from the same sample excluded from the training set for when they were being used as a test set? Otherwise, you are training and testing on the same samples. It is not clear how you are doing the cross-validation.

We are thankful for pointing out the missing part. The dataset was split in half for the training and testing set (line 390), but we did not mention it earlier. We updated line 242 to reflect this part.

Line 240: How was this done?

We clarify (line 257) that the preliminary investigation was done with GC-MS, and the concentration was roughly estimated with a non-validated method using a non-validated headspace GC-MS analysis with random EtOH calibrators.

Line 271: Was this human urine or artificial?

Human urine was used as mentioned in line 194.

Line 323: Figure 3 is really difficult to read and there doesn’t appear to be any axes labels on the figure (though it might be the poor quality of the image). I wonder if the PDF process has caused this?

We believe the PDF process degraded the quality of our images, but we have updated the figure with a higher resolution image. We hope the quality is maintained after PDF processing.

Line 358: I am pretty sure there are some UTI studies using direct analysis. The authors should comment on these papers as well.

This is a useful suggestion. We included a new reference in the introduction. Aathithan et al. direct urine analysis but they use a simple PCA to detect either a positive or negative infection.

Line 397: Why is this important? Would not the sample be tested and then disposed of? If a second reading was needed, they would just take some more out of the sample container – or just get more urine from patient. The reason for doing this needs to be explained. You are providing evidence that you should just test once. Line 415: Why would you measure for 30 minutes with an electronic nose? For what purpose?

We appreciate the valuable suggestion. We hope that we have clarified the confusion on measurement time in response #4 above. Briefly, a longer measurement time can expand the number of examples collected as long as sample quality is maintained. Here, we collected data for training or testing the model three times (10 minutes each). We showed that beyond three times, the samples started to degrade, and these data should not be included. This way, we can maximize the inclusion of useful data without wasting samples (higher efficiency).

Line 427: There have been a number of studies looking at urine stability with electronic noses – which I noticed are not referenced. Also, the result found here is well known in the electronic nose community.

We revised our manuscript (lines 461) to include a paragraph discussing sample instability in general that is due to storage time and initial measurement with two new references (Ref. 23 & 24). We also referenced Roine et al. (Ref. 6) for their speculation that a short measurement time already allows for a good classification. However, we reason that this can only be made after the fact by classification results and that number of examples is a trade-off for shortening measurement time. Here, we devised a method to determine the optimal timepoint within which data can still be taken instead of a retroactive determination. We hope the urine stability has been well discussed and referenced accordingly. However, please let us know if the discussion needs further improvement and more references are needed. Although the results found here is well known in the electronic nose community, but the use of validated Headspace GCMS method to establish and optimise electronic nose measurement can be considered novel and the results from this study shows the usefulness of the headspace GCMS method in optimisation and quality control of ENose measurement.

Line 437: This is really important – and much more that the focus on electronic noses. I would like this to have been in the introduction.

We updated to have this line in the introduction (line 137). We did highlight the importance of sample instability and the need for the GC-MS method to determine the cut-off for measurement time in the introduction.

Line 445: Would this not be dependent on the level of infection in real life?

Indeed, the level of infection could be proportional to the bacterial concentration and directly affect VOC concentration. In this study, we used different strains from the same species and still achieved good classification.

Reviewer #3: 

1. The capabilities of quantifying individual VOCs is not normally an important requirement for identification of diseases or pathogens responsible for causing diseases when using electronic-nose devices. The most important information to validate is the identities of VOCs making up the E-nose smellprint signatures and thus VOC profiles, not concentration of VOCs which is normally only needed in metabolomic studies to determine effects of pathogens on metabolic pathways of the host. Thus, quantification does not add significantly to the capabilities for testing the efficacy of a new experimental e-nose device. The ultimate objective was to develop the capabilities of the e-nose for detecting UTI caused by different microbes. The methods developed here do not contribute to that objective and the data obtained is normally part of a pilot study for methods development and not published as a stand alone unit without e-nose data on UTI samples from different types of microbial causes with adequate controls. Quantification of a single possible VOC in a UTI sample headspace provides very little information towards development of an e-nose library database containing specific complex mixtures of VOCs that affect the output signatures of the e-nose sensor array. The sensor array responds to all of the VOCs present in the headspace, not just a single VOC such as ethanol.

We appreciate the critique raised and fully understand that e-noses rely on the fingerprint of VOC profile and not necessarily concentration. However, we quantified ethanol in order to understand how sample quality is affected by measurement events and how we could maximize data collection while maintaining their quality in order to obtain a robust model as efficiently as possible. The key takeaway should be that by simply determining the proper cut-off time for data collection, we can avoid taking in unqualified data that potentially poison the model. We devised a working method for making that determination by using headspace GC-MS and disseminated a direction for future data control that helps enhance the classification accuracy.

2. All of the figures (Fig. 1-7) are of very low quality resolution and do not provide the data necessary to support the efficacy of a new experimental e-nose device for UTI diagnostics (based on quantitation of EtOH alone). Development of methods for quantitation of EtOH along with a standard curve alone do not contribute significantly towards development of e-nose methods useful for UTI diagnostics and thus the Conclusions are not supported by the objectives of the study or the data obtained towards this purpose.

We uploaded new figures of higher resolution. We agree that the GS-MS method does not directly contribute to the development of a better e-nose. However, improvement of the electronic nose involves many aspects, including choice of materials, of algorithm, and of data quality. 

We explained in the introduction our rationale for choosing MOX sensors and an artificial neural network to create a potentially more powerful e-nose. We clarify in the introduction that the objectives of the study is to demonstrate the discrimination between K12 and UPEC E. coli using the EVA, which we achieved. Furthermore, ethanol concentration can be indicative of sample stability. After determining the timepoint at which the concentration deviates beyond the acceptable range, we could limit our data collection within this time frame and improve the classification model independently of either hardware (sensor choice) or software (classifier choice). 

Reviewer #4: 

1. The paper describes the use of an electronic nose to distinguish the pathogens causing urinary tract infection, and proposes an new experimental method to improve sample stability during e-nose measurements. The problem of sample stability during e-nose measurment should be better claimed in the introduction, because it is the focus of the described sperimentation.

We appreciate Reviewer 4 suggestion. We would like to clarify that sample stability was not improved in this study, but rather data quality was improved by terminating data collection before the samples became unstable. As suggested, we revised the introduction to better highlight the problem of sample degradation over time.

2. The methods involved to prepare EVA samples and GC-MS calibrants are described properly. However, the authors should clearly state the reasons leading to the use of normal urine samples to be inoculated rather than real infected samples. 

We agree with this suggestion and revise line 379 to describe the rationale for using our method. In this paper, we devised a benchtop model of UTI by infecting the normal urine samples because it is easy to access, establish, and control in terms of bacterial culture (K12 vs UPEC) and ethanol concentration. Using real infected samples would be valuable for validating the e-nose. However, patient-derived samples could have extremely varied bacterial profiles as well as VOC profiles. Since the goal is not to simply detect between infected and non-infected, we have to carefully control the sample composition to claim the discriminating power of the e-nose. By using a benchtop model, we have decoupled confounding factors to gain a better understanding of how data quality can contribute to the robustness of our e-nose approach.

3. I think that authors should also better describe the approach involved for training the e-nose. Specifically, they should describe the scheme involved for presenting samples to the e-nose and validating the classification performance. The paragraph (lines 223-231) is not clear. The value of the sensor resistance at different times was used as features?

We included a new SI figure (S3) to show the setup of samples with the e-nose. Our feature extraction was achieved through a amplitude-driven approach of extracting the mean area under the curve for the given response duration. To focus the scope of the paper on the GC-MS validation, we did not go into much detail on this approach, however more details on this approach and the reasoning for its application can be found in our patent-filing, “Feature tuning – application dependent feature type selection for improved classification accuracy.” https://patents.google.com/patent/US20210172919A1/en

Following the preprocessing stage to extract salient features, we used a simple feed-forward multilayer-perceptron ANN consisting of three layers for classification, using the opensource GNU Octave -v5.1.0.0 (GUI).

4. All figures of the paper have a very poor quality. They should be revised. In some cases, the text is not readable. Moreover, the authors should add reference supporting their statements throughout the paper.

We revised with figures of higher resolution and more references. Please view figures individually to maintain integrity of resolution.

---

## [Decision Letter · Decision Letter 1]

18 Jun 2022

PONE-D-21-22287R1Quantitation of ethanol in UTI assay for volatile organic compound detection by electronic nose using the validated headspace GC-MS methodPLOS ONE

Dear Dr. Adebiyi,

Thank you for submitting your manuscript to PLOS ONE. After careful consideration, we feel that it has merit but does not fully meet PLOS ONE’s publication criteria as it currently stands. Therefore, we invite you to submit a revised version of the manuscript that addresses the points raised during the review process.

ACADEMIC EDITOR: Still reviewer # 2 is raising a major concern over the revised form of the MS. Would you please go through the comments and amend the MS accordingly.  

We look forward to receiving your revised manuscript.

Kind regards,

A. M. Abd El-Aty

Academic Editor

PLOS ONE

Reviewers' comments:

Reviewer's Responses to Questions

**Comments to the Author**

1. If the authors have adequately addressed your comments raised in a previous round of review and you feel that this manuscript is now acceptable for publication, you may indicate that here to bypass the “Comments to the Author” section, enter your conflict of interest statement in the “Confidential to Editor” section, and submit your "Accept" recommendation.

Reviewer #1: All comments have been addressed

Reviewer #2: (No Response)

Reviewer #5: All comments have been addressed

2. Is the manuscript technically sound, and do the data support the conclusions?

Reviewer #1: Yes

Reviewer #2: Partly

Reviewer #5: Yes

3. Has the statistical analysis been performed appropriately and rigorously? 

Reviewer #1: Yes

Reviewer #2: Yes

Reviewer #5: (No Response)

4. Have the authors made all data underlying the findings in their manuscript fully available?

Reviewer #1: Yes

Reviewer #2: Yes

Reviewer #5: Yes

5. Is the manuscript presented in an intelligible fashion and written in standard English?

Reviewer #1: Yes

Reviewer #2: Yes

Reviewer #5: No

6. Review Comments to the Author

Reviewer #1: I was excited to see your new version.

All my previous comments have been addressed.

The changes to the introduction made it more relevant and interesting to read.

On the current version of the manuscript I have noticed some grammar/editing points:

- Line 143 "in" should be added: "results (in) under two minutes".

- Line 303 there are two full stops after mL/min.

- Line 370 would be better if "there are" was used : "[...] (there are) numerous studies on the e-nose application [...]".

I hope your future research projects will go well. I am looking forward to seeing more IBM EVA use.

Reviewer #2: The paper is much improved, but there are a number of points outstanding and a number of answers were not added to the manuscript. Below are some more specific comments.

Line 97: I would add a reference to the original paper on this by Dodd & Persaud in Nature from the early 1980s.

Line 106: I am surprised there haven’t been more papers and more recent ones. These are very old. Can this table be updated?

Line 113: PCA is not a classifier, so this will need to be altered to make sense.

Line 133: I would add the paper by Estfani or urine storage by FAIMS (I might have the name wrong). It showed samples were good for 9months+ at -80C. I am also sure that some of Dutch group have done work on storage at room temperature. Might be worth adding an extra reference or two here.

Line 153: Usually ethanol is undertaken using a single optical gas sensor – for example they are used in breath ethanol testing. This should be included in the description as they are pretty common.

Line 162: I am pretty sure the EVA has been reported before. I would rephase this sentence.

Line 181: So, why not just use the ethanol sensor are be done?

Line 461: I am more interested in the stability of your instrument and the sensors over the sample. I may not have got to it yet, but how long did the calibration last?

Line 512: I would like to see a section on limitations of the study and of the use of eNoses for this purpose. For example, drift (of the various forms) is a good example. Also, how well the system will be able to cope with variations in urine. With UTIs urine can be almost clear and others the consistency of porridge.

I would still like to see a PCA from this dataset in the paper and a loading plot for the PCA. Unless the authors are saying this is the same dataset, then adding a new PCA is appropriate.

The length per test is still significant. It just needs to be added as a limitation of the study as there is no way to re-do the experiments. You can get electronic noses to respond in seconds, just not how you tested it.

From the previous points (and I state above) I can just go purchase a cheap ethanol sensor, so the reason for needing to use an electronic nose for this purpose needs to be explained in more detail. I could even get a cheap MOX sensor and make it a lot more specific to ethanol…you just need to justify better why you would use an expensive eNose over a cheap single sensor.

Details of the EVA need to make it to the paper (I didn’t see the additions). Also add details of the optimisation and not just the patent – to the paper…

Please add all feature extraction methods to the paper. Also software used etc.

Reviewer #5: Though the manuscript is well designed and presented, I am doubtful regarding the applicability of the developed method. The manuscript can be considered for publication provided that comments and suggestions given by the other reviewers are fully addressed.

7. PLOS authors have the option to publish the peer review history of their article (what does this mean?). If published, this will include your full peer review and any attached files.

Reviewer #1: No

Reviewer #2: No

Reviewer #5: No

---

## [Author Response · Author response to Decision Letter 1]

30 Jul 2022

RESPONSE TO REVIEWERS (PONE-D-21-22287)

We are delighted to receive positive comments from reviewers # 1 and 5 and acknowledge concerns raised by reviewer #2. We have addressed all the comments in our revised manuscript, and the response details are as follows. We are happy to answer further concerns and questions from the reviewers.

Reviewer 1:

"I was excited to see your new version. All my previous comments have been addressed. The changes to the introduction made it more relevant and interesting to read. I hope your future research projects will go well. I am looking forward to seeing more IBM EVA use."

We appreciate all the positive comments from Reviewer 1 and grateful for your careful evaluation of our manuscript.

Line 143 "in" should be added: "results (in) under two minutes"

Please note that the line numbers hereafter refer to the no-mark-up version of our new revision.

Line 148: We corrected it. 

Line 303 there are two full stops after mL/min.

I could not identify the mistake. The manuscript had only one full stop after mL/min at line 301.

Line 370 would be better if "there are" was used : "[...] (there are) numerous studies on the e-nose application [...]".

Thank you for your suggestion. We made the change at line 379.

 

Reviewer 2

"The paper is much improved, but there are a number of points outstanding and a number of answers were not added to the manuscript. Below are some more specific comments." 

Line 97: I would add a reference to the original paper on this by Dodd & Persaud in Nature from the early 1980s.

Please note that the line numbers hereafter refer to the no-mark-up version of our new revision. 

Line 87: We added the paper, which was an excellent reference to the origin of e-nose technology.

Line 106: I am surprised there haven't been more papers and more recent ones. These are very old. Can this table be updated?

Line 95: We updated the table with a few more recent references. However, to our surprise, not many new papers specifically looked at the urine-based classification of pathogens in the context of urinary disease. We reflected this in the paragraphs following table 1. 

Line 113: PCA is not a classifier, so this will need to be altered to make sense.

We removed that line and modified the paragraph (not lines 97-108).

Line 133: I would add the paper by Estfani or urine storage by FAIMS (I might have the name wrong). It showed samples were good for 9months+ at -80C. I am also sure that some of Dutch group have done work on storage at room temperature. Might be worth adding an extra reference or two here.

Line 114: We appreciate your helpful comment. Esfahani et al. (2016 & 2018) are good papers to be included as it shows that despite storing at -80oC, storage time still significantly affects classification accuracy. They suggested samples were good within 9 months (in the 2016 paper) and that performing urine analysis within a year of storage resulted in better accuracy than including all samples within four years. Their 2018 study only considered two labels (healthy vs. diabetes). An even shorter storage time could be recommended for a more complex analysis involving more labels.

Line 153: Usually ethanol is undertaken using a single optical gas sensor – for example they are used in breath ethanol testing. This should be included in the description as they are pretty common.

Line 138: We updated this line to add the breath analyzer as an example of an ethanol quantification system.

Line 162: I am pretty sure the EVA has been reported before. I would rephase this sentence.

Line 146 We have introduced the EVA before in a conference proceeding (Adebiyi et al. 2019). Therefore, we rewrote the sentence as "In this paper, we used the Electronic Volatile Analyzer (EVA)…" and cited this reference.

Line 181: So, why not just use the ethanol sensor are be done?

Line165: Ethanol sensors alone will not create enough differential response to discriminate many different VOC profiles. We mainly measured ethanol as an indication of sample stability because we inoculated urine with ethanol-producing bacteria (E. coli). However, a robust e-nose requires a combination of sensors responsive to various compounds. We revised this line to reflect these points.

Line 461: I am more interested in the stability of your instrument and the sensors over the sample. I may not have got to it yet, but how long did the calibration last?

Line 434: We believe we shared this in our previous review, but the stability of our sensors over the sample period was reproducible from measurement to measurement. From a cold-start of the instrument, we ran air (our carrier gas) through the chamber for sixty minutes, before we moved on to the measurement samples. Between each 15 minute measurement, we ran air for five minutes to purge any residual VOCs. 

Line 512: I would like to see a section on limitations of the study and of the use of eNoses for this purpose. For example, drift (of the various forms) is a good example. Also, how well the system will be able to cope with variations in urine. With UTIs urine can be almost clear and others the consistency of porridge.

Line 481: We added a paragraph on the limitations of the study. We think the consistency of urine is not a problem per se but what VOC profile they will produce is the determining factor. Obviously, cloudy vs clear urine may have different VOCs in them, some may not be even related to UTI. Therefore, screening for prevalent compounds in UTI vs healthy controls is important and optimizing sensor choice that targets these compounds. Sensor drift is a notable limitation of electronic noses given the variability of the MoX sensors in undergoing these oxidation reactions over time. Sensor drifting can be compensated by a multitude of mathematical techniques (newly included in the paper) such as baseline correction and the like, but our instrument is designed for real-time use”in the wild”, so this choice of calibration was a design choice meant to be as robust as possible to accommodate mostly operator-free scenarios.

I would still like to see a PCA from this dataset in the paper and a loading plot for the PCA. Unless the authors are saying this is the same dataset, then adding a new PCA is appropriate.

Our dataset is the same as the last review. We have tried PCA but found ANNs more suitable to capture the drift and variability in the model for our use-case and setup (which we also described previously). We are happy to provide the reviewer with a PCA for their reference, but a study by itself can be conducted on the choice of mathematical approaches suitable. Given the scope of this paper was mainly to quantify the levels of ethanol in our samples with the validated GC-MS method, we did not go into all those details. An example of a reference related to these mathematical methods with the PCA loadings can be found in Figure 3 of this reference [30] in our paper, which provides these details.

Adebiyi, A., Than, N., Swaminathan, S., Abdi, M., Bowers, A. N., Fasoli, A., ... & Bozano, L. (2020, January). Rapid Strain Differentiation of E. coli-inoculated Urine Using Olfactory-based Smart Sensors. In SEIA'2019 Conference Proceedings (p. 307).

The length per test is still significant. It just needs to be added as a limitation of the study as there is no way to re-do the experiments. You can get electronic noses to respond in seconds, just not how you tested it.

We have addressed it in the limitation paragraph. However, we would like to add that this length (15 minutes) signifies the measurement stage to train the model. At inference, classification takes 80 seconds. 

From the previous points (and I state above) I can just go purchase a cheap ethanol sensor, so the reason for needing to use an electronic nose for this purpose needs to be explained in more detail. I could even get a cheap MOX sensor and make it a lot more specific to ethanol…you just need to justify better why you would use an expensive eNose over a cheap single sensor.

Line 475: We appreciate that cheap ethanol sensors are ubiquitous and finding a correlation between the quantity of ethanol and E. coli strain type could potentially aid in strain differentiation. In some ways, this is what we sought to explore by using the validated GC-MS method to determine if the change in ethanol is mainly due to the degradation of the sample vs the ethanol secreted by the given E. coli strain at incubation. It could be more difficult for a cheap ethanol sensor to tell if it is UPEC E. coli after the sample had been sitting for three days than K12 after two. If we did seek to use the cheap ethanol sensors, we would stil have to validate it, but we could add that as a point. Speaking to the justification of the EVA as a device, we are seeking to build a library of different types of bacteria, not just speficially those that are related to ethanol, it just happens that ethanol is the main VoC in this use-case.

Details of the EVA need to make it to the paper (I didn't see the additions). Also add details of the optimisation and not just the patent – to the paper…

As we mentioned previously, the scope of this paper is intended to look at the quantification of the ethanol method, and not to provide a deep dive into the hardware optimization of the EVA, which could be its own study. We are happy to reference this patent in paper, but believe that adding all the details will take away from the focus of this study. However, we briefly summarize the operational basis of EVA on page 10, which we believe is appropriate to reflect the focus of this paper.

Please add all feature extraction methods to the paper. Also software used etc.

Line 247: We have added this to the paper.  

Reviewer #5: 

Though the manuscript is well designed and presented, I am doubtful regarding the applicability of the developed method. The manuscript can be considered for publication provided that comments and suggestions given by the other reviewers are fully addressed.

We are thankful for the positive comment. We believe other researchers in the field can adapt the developed method to characterize the content in their e-nose samples. Not only for ethanol but other compounds can also be quantified by modifying the validated method.

---

## [Editor Report · Decision Letter 2]

20 Sep 2022

Quantitation of ethanol in UTI assay for volatile organic compound detection by electronic nose using the validated headspace GC-MS method

PONE-D-21-22287R2

Dear Dr. Adebiyi,

We’re pleased to inform you that your manuscript has been judged scientifically suitable for publication and will be formally accepted for publication once it meets all outstanding technical requirements.

Kind regards,

A. M. Abd El-Aty

Academic Editor

PLOS ONE

Additional Editor Comments (optional):

The authors respond satisfactorily to the comments raised by the reviewer.
---

## [Editor Report · Acceptance letter]

27 Sep 2022

PONE-D-21-22287R2 

Quantitation of ethanol in UTI assay for volatile organic compound detection by electronic nose using the validated headspace GC-MS method 

Dear Dr. Adebiyi:

I'm pleased to inform you that your manuscript has been deemed suitable for publication in PLOS ONE. Congratulations! Your manuscript is now with our production department. 

Kind regards, 

on behalf of

Prof. A. M. Abd El-Aty 

Academic Editor

PLOS ONE